



# Impact of model structure on flow simulation and hydrological realism: from lumped to semi-distributed approach.

Garavaglia Federico[1], Le Lay Matthieu[1], Gottardi Fréderic[1], Garçon Rémy[1], Gailhard Joël[1],
Paquet Emmanuel[1], and Mathevet Thibault[1]

[1]EDF-DTG, 21 avenue de l'Europe, BP 41, 38040 Grenoble cedex 09

*Correspondence to:* Garavaglia Federico (federico.garavaglia@edf.fr)

**Abstract.** Model intercomparison experiments are widely used to investigate and improve hydrological model performances. However, a study based only on runoff simulation is not sufficient to discriminate different model structures. Hence, there is a need to improve hydrological models for specific signatures of streamflow (e.g. low and high flow) and multivariable predictions (e.g. soil moisture, snow and groundwater). This study assesses the impact of model structure on flow simulation and hydrological realism using three versions of an hydrological model called MORDOR: the historical lumped structure and a revisited formulation inflected for lumped and semi-distributed structures. In particular, the main goal of this paper is to investigate the relative impact of model equations and spatial discretization on flow simulation, snowpack representation and evapotranspiration estimate. The models comparison is based on an extensive dataset composed of 50 catchments located in French mountainous regions. The evaluation framework is founded on a multi-criteria split sample strategy. All models were calibrated using an automatic optimization method based on an efficient genetic algorithm. The evaluation framework is enriched by the assessment of snow and evapotranspiration modeling against in-situ and satellite data. The results showed that the new model formulations perform significantly better than the initial one in terms of the various streamflow signatures, snow and evapotranspiration predictions. The semi-distributed approach provides better calibration-validation performances for snow cover area, snow water equivalent and runoff simulation especially for nival catchments.

## 1 Introduction

Hydrological models are widely applied in water engineering for design and scenario impact investigations. Depending on the type of application, the catchment characteristics or the data availability, different model conceptualizations and parameterizations are considered. In many cases, choice of the model is the result of the modeller's experience. However, hydrologists have developed objective and rigorous frameworks to evaluate and improve hydrological models.

A common approach to discriminate different model structures is to perform model intercomparison experiments. Such experiments have been helpful to explore model simulation performance of lumped (e.g., Duan et al., 2006; Breuer et al., 2009), semi-ditributed (e.g., Duan et al., 2006; Holländer et al., 2009) and distributed (e.g., Henderson-Sellers et al., 1993; Reed et al., 2004; Holländer et al., 2009; Smith et al., 2012) models in a consistent way using the same input data. To overpass hydrological singularity and provide general conclusions, multi-catchments experiments have been proposed by several authors



(e.g. Perrin et al., 2001; Gupta et al., 2014) and are now extensively used. Most of studies focus only on runoff modelling performance, since runoff is the main data available at catchment scale. However, as the runoff data is used for both the model training and its validation, one may question the quality of prognostic variables produced by the model which have not been optimized through calibration, such as snow, evapotranspiration and soil groundwater. Moreover, when focusing only on runoff

simulation, we often fail to discriminate different model structure. On the contrary, interesting conclusions may be drawn when focusing on particular aspects of streamflow not used in the calibration process : low flows (Staudinger et al., 2011) or high flows (Vansteenkiste et al., 2014) or on other hydrological variables, such as soil moisture (Orth et al., 2015), snow (Parajka and Blöschl, 2006) or groundwater (Motovilov et al., 1999; Beldring, 2002)).

In a similar way, the aim of this paper is to compare different model structures in terms of both runoff simulation and

hydrological realism. More precisely, we investigate the relative importance of model equations and spatial discretization on flow simulation, snowpack representation and evapotranspiration estimate. Such correspondence between model and "reality," often described as "working for the right reasons" (Kirchner, 2006; Kavetski and Fenicia, 2011; Euser et al., 2013), is essential if the model is to be used as a tool for improving the understanding of a hydrological system, and/or used for prediction and extrapolation, such as simulating the impacts of land use change, variability in climatological forcing, etc.

We apply this framework on MORDOR hydrological model (Garçon, 1996), which has been extensively used in Électricité de France (EDF, French electric utility company) for more than 25 years for operational applications. Recent changes in the model structure have been realized in order to improve model performances. Many alternative model structures have been tested, which concern both model equations and model spatial discretization, and we selected the two best solutions. In this study we present and compare these three new formulations with the historical version.

The paper is organized as follows: The dataset is introduced in Sect. 2. Sect. 3 presents the methods used for this study. We describe the different model structures considered in the comparison, and we present the evaluation strategy. Results of the study are discussed in Sect. 4, before drawing some conclusions and discussing potential future developments in Sect. 5.

## 2  Data and study area

The comparison of the three hydrological models is based on an extensive dataset composed of data from 50 catchments.

This dataset collects different operational case studies from EDF activities. These catchments are located in mountain regions, manly in Alps (18 catchments), Pyrenees (5 catchments) and Massif Central (21 catchments). One catchment is located in the north of France (Ardennes region), one in the north-west (Brittany region) and one in Corsica island. Figure 1 shows the catchment locations. Catchments were chosen based on quality and length of records criteria. The large hydro-climatic range of the data-set allows to ensure the models consistency in different hydrological conditions. By way of example, the average

area at all these catchments is 922 km², ranging from 18.5 to 7401 km² and the average of median elevation of whole dataset is 1046 m a.s.l., ranging from 92 to 2355 m a.s.l.

For each catchment the following data were collected : (i) discharge, (ii) rainfall, (iii) temperature, (iv) reference and actual evapotranspiration, (v) fractional snow cover and local snow water equivalent.





**Table 1.** Main components of MORDOR V0, V1 and SD models in term of water-balance, runoff production, snow model, routing scheme and spatialisation. The number of free parameters is underlined.

| Module | MORDOR V0 | MORDOR V1 | MORDOR SD |
|---|---|---|---|
| Water-balance | Adjusted PET from a statistical formulation driven by temperature. 2 free parameters | Forced by PET. Crop coefficient formulation. 2 free parameters. | |
| Runoff production | 4 storage and 3 flux components (surface, sub-surface and base flows). Linear inflow and outflow of storage. 7 free parameters. | 4 storage and 3 flux components (surface, subsurface and base flows). Non-linear inflow and outflow of storage. 7 free parameters. | |
| Snow model | Snow accumulation driven by the air temperature and hypsometric curve. Classical degree-days formulation for snow melt. 11 free parameters. | Snow accumulation driven by air temperature and parametric S-shaped curve. For snow melt : classical degree-days, cold content, liquid water content, ground-melt component and variable melting coefficient. 6 free parameters. | |
| Routing scheme | UH modeled by Weibull distribution. 2 free parameters. | UH modeled by diffusive wave. 2 free parameters | |
| Spatialisation | None | None | Orographic gradients. 2 free parameters. |
| Total | 22 free parameters | 17 free parameters | 19 free parameters |

The discharge data are provided by EDF and French water management agencies networks. The average length of records at all these stations combined is around 25 years, ranging from 9 years for Ouveze at Bedarrides (South Alps) to 53 years for Sioule at Fades (Massif Central). In total the whole discharge dataset consist of 1526 hydrologic years. The average runoff for the whole dataset is around 803 mm/year, ranging from 225 to 1635 mm/year. In regard to forcing data, rainfall and temperature are

5  gridded and provided by Gottardi et al. (2012). These data result from a statistical reanalysis based on ground network data and weather patterns (Garavaglia et al., 2010). They are available for the 1948-2012 period at 1-km² / 1-day resolution. Concerning the rainfall, the average amount for the whole dataset is around 1345 mm/year, ranging from 825 mm/year to 2000 mm/year. The model time step differs from catchment to catchment and depends on hydrological characteristics (area, topography, time to peak, etc.). We model 44 catchments at daily time step, 1 at 12-hours time step, 2 at 8-hours and 3 and 6-hours time step. In

10  order to obtain forcing at infra-daily time step, the gridded data are downscaled according to ground network data at finer time step, i.e. the shape of locals gauges are used to computed areal precipitation and temperature at 12-, 8- and 6-hours time step.

Evapotranspiration data used for validation come from the satellite MOD16 global evapotranspiration product (Mu et al., 2011), which provides 1-km² / 8-day land surface ET datasets since 2000 using Penman-Monteith equation and a surface resistance derived from MODIS surface data. Compared to local flux measurements, MOD16 product has the great advantage





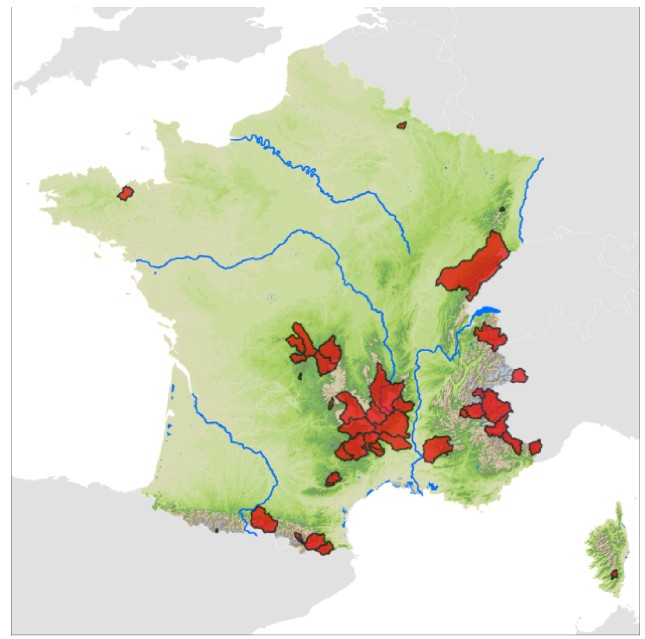

**Figure 1.** Localization of studied catchments

to provide spatially explicit large scales ET estimates. Some studies have shown its consistency, even if it is known for being plagued by many uncertainties, especially in mountainous areas where the global meteorological input is clearly deficient or in the tropics and subtropics regions where it clearly underestimates ET (e.g. Trambauer et al. (2014), Hu et al. (2015), Miralles et al. (2015)). Two types of snow data are used for model validation: fractional snow cover (FSC) and snow water equivalent

(SWE). The satellite MOD10 product (Hall et al., 2002) provides us fractional snow cover time-series at catchment scale. This product is available since 2000 at a 500-m / 1-day resolution, and is widely used for hydrological applications (e.g. Rodell and Houser (2004), Parajka and Blöschl (2006) or Thirel et al. (2013)). Snow Water Equivalent (SWE) data come from EDF snow network, composed by Cosmic-Ray Snow Sensors (NRC) (Kodama et al., 1979; Paquet and Laval, 2006). In this study we use three measurement gauges situated within the Durance at Clapiere catchment : Izoard (2280 m a.s.l), Chardonnet (2455 m

a.s.l.) and Marrous (2730 m a.s.l.). SWE time-series at these locations are available since 2001.

## 3  Methods

### 3.1  Hydrological model versions

#### 3.1.1  Initial model formulation (MORDOR V0)

The historical MORDOR model is a lumped conceptual rainfall–runoff model. Its structure is similar to that of many conceptual

type models, with different interconnected storage. It works continuously, and can be used with a time step ranging from hourly





to daily. Required input data are mean areal rainfall and air temperature.

The main components of the model are: (i) an evaporation function that determines the potential evaporation as a function of the air temperature; (ii) a rainfall excess/soil moisture accounting storage $U$ that contributes to the actual evaporation and to the direct runoff; (iii) an evaporating storage $Z$, filled by a part of the indirect runoff component, that contribute to the actual
evaporation; (iv) an intermediate storage $L$ that determines the partitioning between a direct runoff, an indirect runoff and the percolation to a deep storage $N$; (v) a deep storage $N$ that determines a baseflow component; (vi) a snow accumulation function calculated from the temperature and the hypsometric curve of the catchment and a rain-snow transition curve; (vii) a snow melt function based on an improved degree-day formulation; (viii) a unit hydrograph that determines the routing of the total runoff.

In this configuration, the MORDOR V0 model has 22 free parameters (see Table 1) to be optimized during the calibration process. The model was developed in the early 1990s (Garçon, 1996). Since then, it has been extensively used at EDF for operational inflow and long-term water resources forecasting, hydrological analysis and extreme flood estimation (Paquet et al., 2013). Few hundred models have been calibrated in France and abroad (Mathevet and Garçon, 2010). A model inter-comparison study (Mathevet, 2005; Chahinian et al., 2006) based on the assessment of 20 rainfall–runoff models, tested on a
sample of 313 catchments at the daily and hourly time-step, has shown that MORDOR model is among the more efficient and robust rainfall–runoff model structures. Valéry et al. (2014) also showed that MORDOR snow module was among the most efficient when compared to 6 well-known snow modules.

    However, some motivations for model improvement have appeared recently: (i) increase of model performances in term of floods and low-flows simulations may widen model applications; (ii) snow modelling have to be improved for allowing snow
data assimilation, specially for long-term snow melt forecasts; (iii) representation of orographic meteorological variability should be taken into account; (iv) simplification of model structure and parameterization may improve the efficiency of model calibration and reduce parameter equifinality (Beven and Freer, 2001).

### 3.1.2   Proposition of a revisited formulation (MORDOR V1)

The revisited model formulation, hereafter called MORDOR V1, does not modify the overall catchment conceptualization.
In the following parts, we distinguish changes in : (i) the water balance formulation; (ii) the runoff production; (iii) the snow model; (iv) the routing scheme. A special focus on MORDOR V1 components and fluxes is given in Appendix A. In this configuration, the MORDOR V1 model has 17 free parameters to be optimized during the calibration process (see Table 1).

### Water-balance

The water-balance formulation includes a simplified vegetation component, with a maximal evaporation that is derived from the reference evapotranspiration $ET_0$ using a crop coefficient (Allen et al., 1998). From the maximal evapotranspiration, model calculates actual evapotranspiration (AET) from three components: (i) a surface interception: the precipitation is firstly neutral-ized by the maximal evapotranspiration (e.g. Perrin et al. (2003)); (ii) an evapotranspiration from the root soil, calculated as a linear function of the saturation level of the soil moisture accounting storage $U$; (iii) an evapotranspiration from the capillarity



water storage in the hillslope, calculated as a linear function of the saturation level of the capillarity storage $Z$.

**Runoff production**

The model identifies three flux components: (i) surface runoff; (ii) sub-surface exfiltration; (iii) base flow. Surface runoff is
generated by water excess coming from $U$ and $L$ storage. It represents, in a pure conceptual way, both Hortonian and Hewlettian runoff. Sub-surface runoff is generated by $L$ storage outflow, calculated as a non-linear function of the relative saturation. Base flow is generated by $N$ storage outflow, calculated as a non-linear function of the water content.

**Snow and glacier model**

The snow model is derived from a classical degree-day scheme, with a few important additional processes: (i) a cold content able to dynamically control the melting phase; (ii) a liquid water content in the snowpack; (iii) a ground-melt component; (iv) a variable melting coefficient, depending on the potential radiation and supposed to model the changing albedo effect throughout the melting season. The accumulation phase is controlled by the discrimination of the liquid and solid fractions of the precipitations. From the temperature, these fractions are derived from a classical parametric s-shaped curve (e.g., Zanotti
et al., 2004; Micovic and Quick, 1999). The snowpack is represented by three state variables: (i) the snow water equivalent; (ii) the snowpack bulk temperature; the liquid water content in the snowpack. Snow melt is calculated as the sum of superficial and ground melts. Superficial melt is derived from a degree-day formulation, where the melting temperature is snowpack bulk temperature, updated at each time-step thanks to air temperature. A glacier component may also be activated. It relies on a simple degree-day formulation.

**Routing scheme**

The transfer function is applied on the sum of the runoff contributions. Its formulation is based on the diffusive wave equation (Hayami, 1951).

### 3.1.3 Semi-distributed model formulation (MORDOR SD)

Semi-Distributed MORDOR model is an improvement of MORDOR V1 model which includes a spatial discretisation scheme. This discretisation is based on an elevation zones approach, which is known to be both parsimonious and efficient for mountainous hydrology (Bergstroem, 1975; DHI, 2009). A special focus on MORDOR SD components and fluxes is given in Appendix A. Figure 2 illustrates such a discretisation on the Durance at La Clapière catchment (2175 km², south Alps), with 10 zones each representing between 5% and 18% of the total area. In most of MORDOR SD applications, spatial variability of mete-
orological forcings is resumed with two orographic gradients, one for precipitation and one for temperature. In this way, we assume that in mountainous areas, the spatial variability is mainly driven by altitude. Most of the model state variables are calculated for each elevation zone. Only groundwater water-content and outflow are considered as global and are calculated at the catchment scale. In the configuration used in this study, the MORDOR SD has 19 free parameters (i.e. 17+2 with 2 orographic gradients) to optimize during the calibration process.




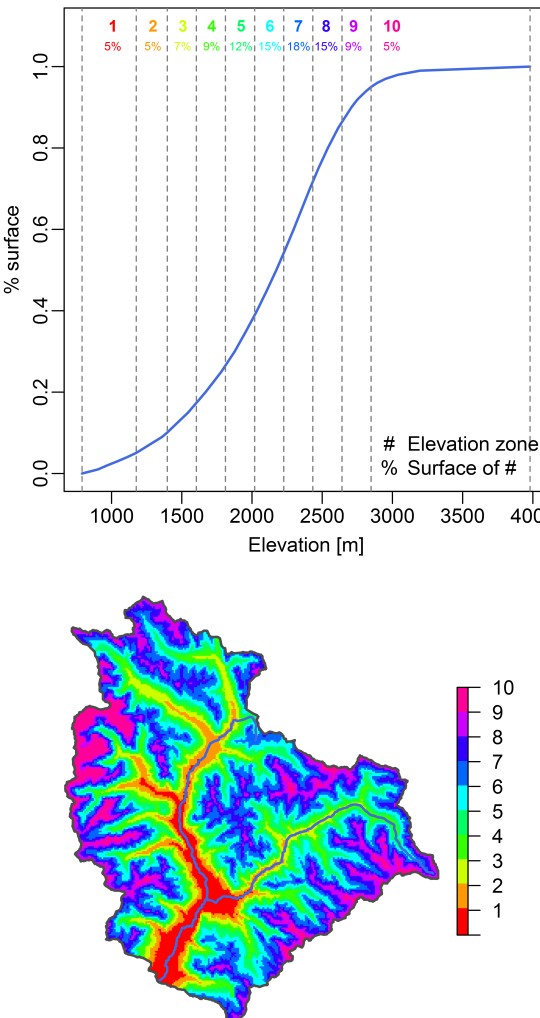

**Figure 2.** Durance@Clapiere catchment: (a) hypsometric curve; (b) elevation zones.

## 3.2 Evaluation strategy

### 3.2.1 Split sample test

For model evaluation, we adopted the split sample test advised by Klemeš (1986). For each catchment, the entire data record was split into two periods (P1 and P2). In the tests, we first calibrated the models on period P1 and tested them in validation mode on period P2. Then the role of the periods was reversed (calibration on P2 and validation on P1). Therefore a total of 100





calibrations (50 for P1 and 50 for P2) and 100 validation tests were run on the whole catchment set, and results were analyzed on this basis.

### 3.2.2 Evaluation criteria

The runoff signatures are viewed in such a way that streamflow data may be broken up into several samples, each of them a manifestation of catchment functioning (Westerberg and McMillan, 2015). Thus, a numeric criterion is calculated over five different sample of observations: (i) entire streamflow time-series is obviously the first signature which has to be reproduced by the model (hereafter called $q$); (ii) streamflow interannual daily regime is used to focus on the capacity to reproduce seasonal variation of observations (hereafter called $reg$); (iii) streamflow empirical cumulative distribution allows a focus on the capacity to reproduce streamflow variance and extremes (hereafter called $qcl$); (iv) streamflow recessions focus on low flows (hereafter called $etg$); (v) $1^{st}$-lag streamflow derivate is the last signature focusing on short term variability (hereafter called $dq$). Hereafter, performances results are resumed using the classical Nash-Sutcliffe Efficiency (NSE), computed on each of the five signatures identified above.

### 3.2.3 Calibration technique

The model calibration is performed using an efficient genetic algorithm inspired by Wang (1991). This stochastic population-based search algorithm perform about 40.000 model runs during a classical calibration process.

The multicriteria composite objective function ($OF$) to be minimised during calibration is expressed as follows:

$$OF = (1 - KGE_q) + (1 - KGE_{reg}) + (1 - KGE_{qlc}) \tag{1}$$

where $KGE$ is Kling-Gupta Efficiency (Gupta et al., 2009), which combines three components: correlation, variance bias and mean bias. The triple focus on time-series, interannual regimes and streamflow variances allows to properly identify the different components of the model. Numerous applications of this $OF$, in a large range of hydro-climatic conditions, showed that it was well-purposed for calibration of the MORDOR model.

## 4 Results and discussion

This section presents the results of the model comparison. We focus our attention to the improvement in term of model performance, representation of snow and evapotranspiration processes.

### 4.1 Improvement of model performances

Figure 3 summarizes the model performances of the three model versions over the validation periods. Distributions of NSE values over the 50 catchments (e.g. 100 simulations) are plotted for the five samples of observations described above ($q,reg,qcl,etg$ and $dq$). One can first notice that the three model formulations have good overall performances. NSE(q) values are above 0.8 in validation on more than 80% of the catchments. However, MORDOR V1 and MORDOR SD perform significantly better than



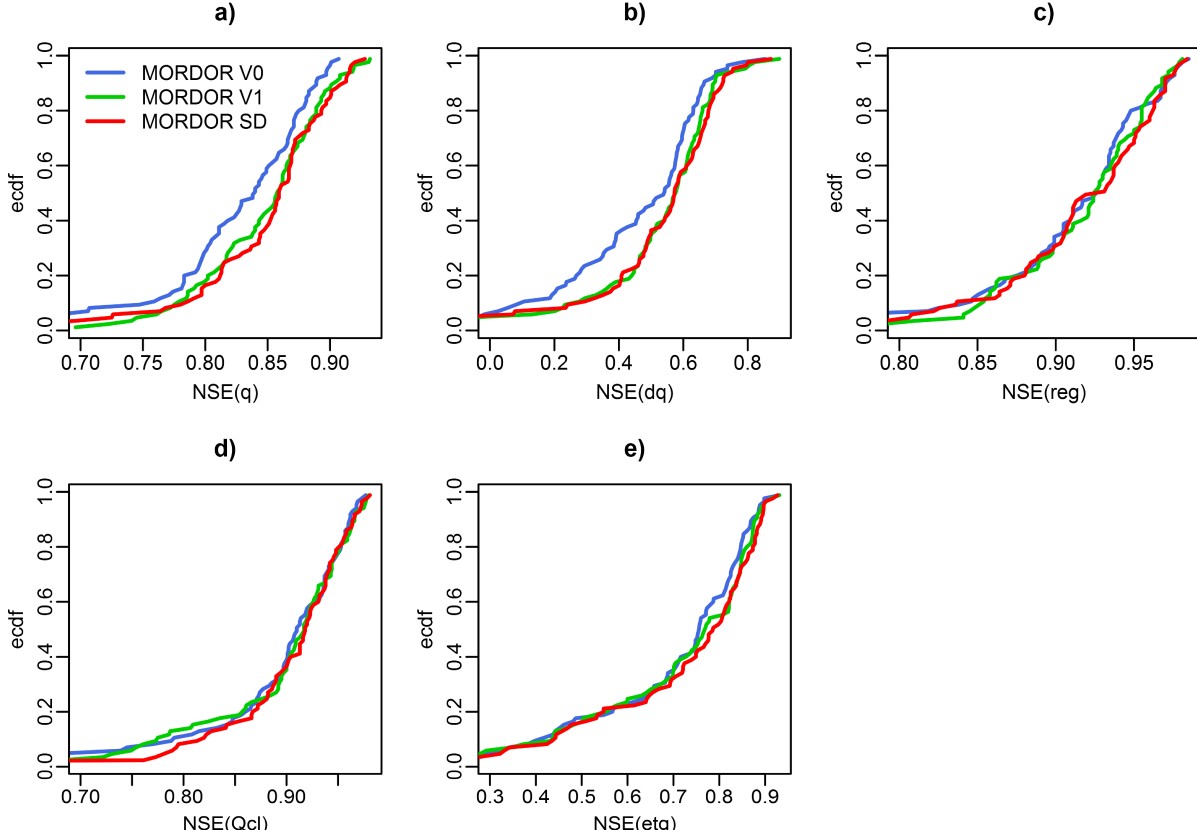

**Figure 3.** Performances of the three model versions on the validation periods, for five streamflows signatures: (a) $q$; (b) $dq$; (c) $reg$; (d) $qcl$; (e) $etg$.

MORDOR V0. This is particularly true for $q$ and $dq$ signatures. This is less significant for $reg$ and $etg$ signatures and insignif-icant for $qcl$. When considering NSE($q$) values, MORDOR V1 and SD have scores above 0.9 for about 10% of the catchments on validation periods. Another interesting result is the very close performance of V1 and SD versions. As a conclusion, the new formulation (V1) provides a spectacular improvement of performances on most of streamflow signatures. In contrast, taking

5   into account orographic meteorological variability has no significant impact on model performances.

    To go further, we compare the mean NSE obtained for each hydrological signature and for the three model versions. In the same time, we distinguish pluvial and nival catchments, according to the classification of Sauquet et al. (2008). Results are illustrated on figure 4. When considering the entire data-set, we confirm previous results: MORDOR V1 and SD have very close performances, which are significantly better than MORDOR V0 ones, especially for $q$, $dq$ and $reg$ signatures

10   (figure 4a). Overall, the relative improvement of performances ranges from 1% to 10%. For the pluvial catchments (figure 4b), conclusions are the same, but overall performances are better. For nival catchments (figure 4c), the picture clearly differs. Overall performances are lower, which underlines the high complexity of processes on these catchments. Moreover MORDOR




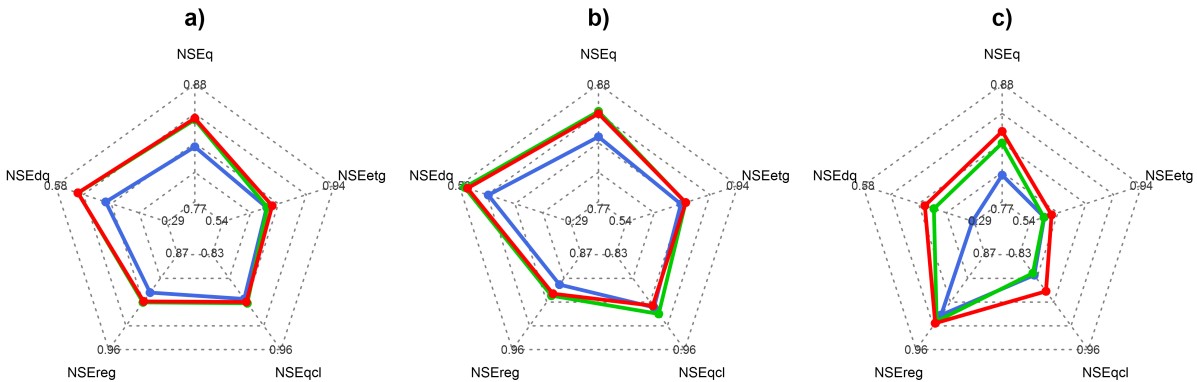

**Figure 4.** Mean NSE for each hydrological signature and for the three model versions: (a) for the entire catchments sample (50 catchments), (b) for the pluvial sample (35 catchments), (c) for the nival sample (15 catchments).

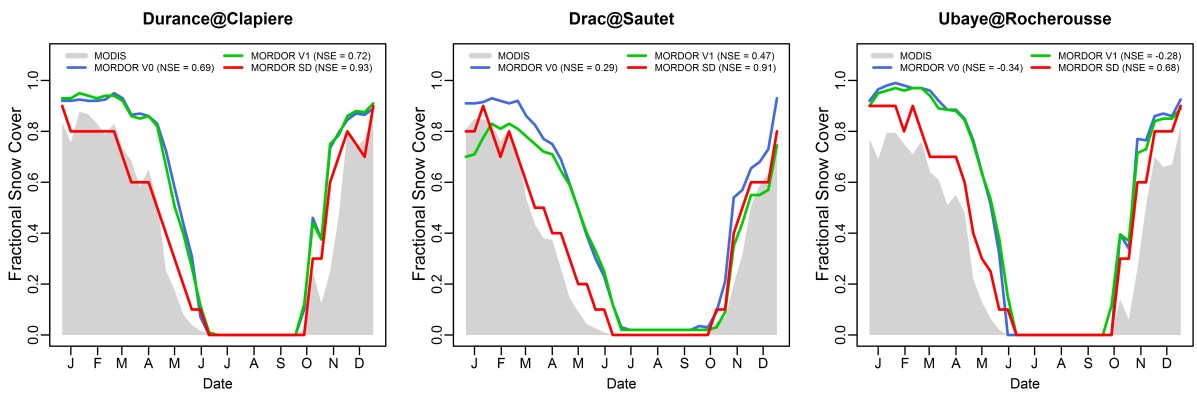

**Figure 5.** Fractional Snow Cover Regime on three mountainous catchments. Comparison of MOD10 SCA product with the three model versions [2000-2012 period].

SD overpasses MORDOR V1 for all signatures. This improvement is especially significant for $qcl$, $q$ and $dq$ signatures, whereas insignificant for $reg$ signature. Therefore, the interest of the semi-distributed scheme clearly appears for nival catchments.

### 4.2 Improvement in the representation of snow processes

One of the objectives of this study was to improve the model representation of snow processes. Hereafter, we investigate this
5   question using two types of data. The first one is a catchment scale aggregation of the fractional snow cover provided by the MOD10 product, available over the 2000-2012 period. The second one is the snow water equivalent at local scale, derived from our NRC observation network.



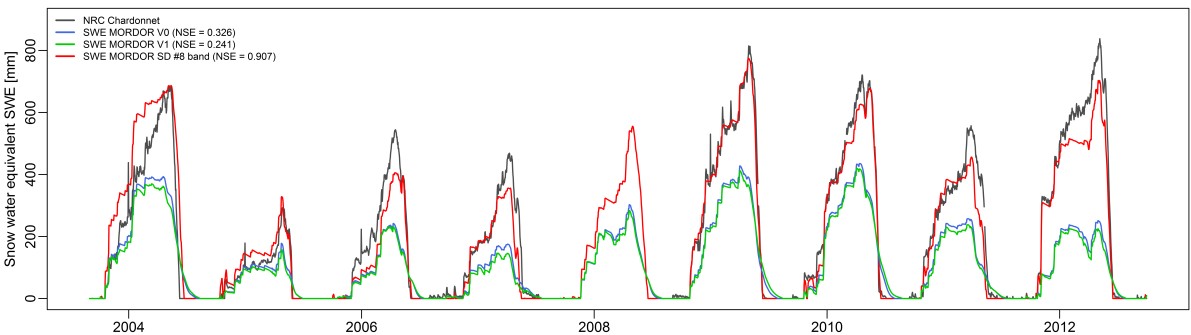

**Figure 6.** Observed and simulated snow water equivalent time-series on the Durance at Clapiere catchment.

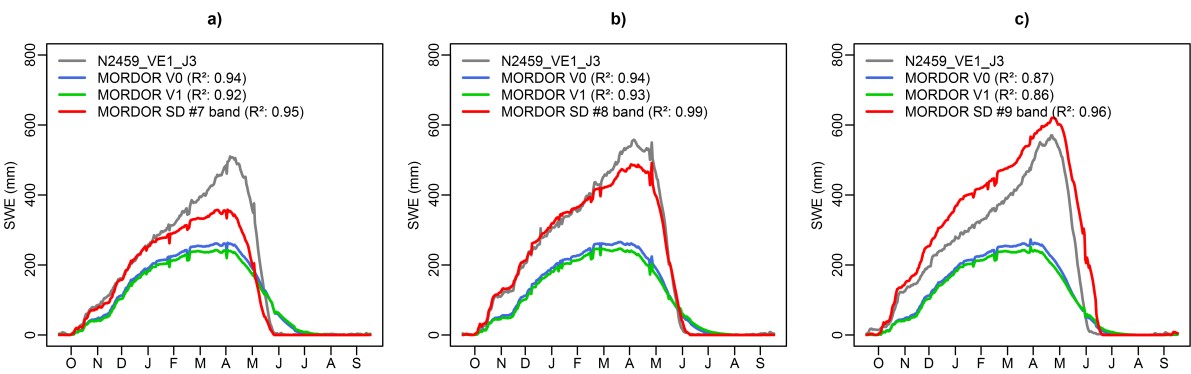

**Figure 7.** Observed and simulated snow water equivalent regimes on the Durance at Clapiere catchment, for three measurement stations : a) Izoard (2280 m a.s.l.), b) Chardonnet (2455 m a.s.l.), c) Marrous (2730 m a.s.l.).

Figure 5 illustrates for three mountainous catchments the regime of modelled and observed fractional snow cover over the 2000-2012 period. MORDOR V0 and V1 show similar behaviour, characterized by a late snow melt and an overestimation of FSC during spring and autumn. On the other hand, MORDOR SD provide a much more realistic FSC on these three catchments, especially during spring. Snowpack discretization within the catchment makes possible to better represent the
5   snow cover evolution. At the end, taking into account orographic meteorological variability allow a significant improvement in FSC simulation, as illustrated by NSE values (see legends of figure 5).

Figure 6 compares observed and simulated SWE time series over the Durance at Clapiere catchment for the 2004-2012 period (observations are missing for year 2008). Observations come from the Chardonnet NRC (2500 m a.s.l.). The MORDOR V0 and V1 simulations (blue and green curves) correspond to the global SWE at the catchment scale, as they do not represent
10   spatial variability. MORDOR SD simulation (red curve) corresponds to the elevation zone #8 situated close to NRC altitude.





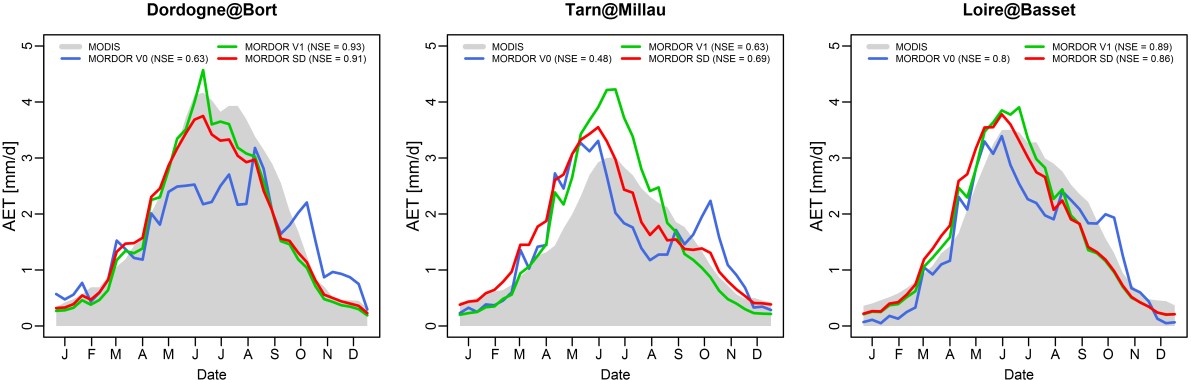

**Figure 8.** Actual evapotranspiration regime (2000-2005) on three pluvial catchments. Comparison of MOD16 AET product with the three model versions.

First, MORDOR V0 and V1 simulations are very similar and significantly underestimate the total amount of SWE. This is a clear conceptual limitation of such global formulations which only simulate bulk values, that can't be compared to local observations. On the contrary, the semi-distributed scheme shows a pretty good agreement when comparing local observations to corresponding elevation zone modelling. MORDOR SD correctly simulates the interannual variability of the maximum

snowpack at this altitude, which varies from about 300 mm in 2005 to about 800 mm in 2007 and 2012. The seasonal dynamic is also very realistic, since both accumulation and melting periods are well simulated. These results are confirmed by Figure 7 for the three snow gauges located over Durance at Clapiere catchment (see Section 2). We compare the observed interannual SWE regime (2000-2012 period) with MORDOR V0,V1 and SD SWE. In figures 7a, 7b and 7c, MORDOR V0 and V1 SWE are respectively the same and corresponds to the bulk SWE at catchment scale. On the contrary MORDOR SD SWE

corresponds respectively to #7, #8 and #9 elevation zones. Logically MORDOR V0 and V1 SWE underestimation increases with elevation. Instead, SWE regimes simulated by MORDOR SD are consistent with at-site observations for all elevations.

### 4.3 Improvement in the representation of evapotranspiration processes

The realism of hydrologic representation is also investigated considering the water-balance, by comparing simulated ET fluxes to MOD16 satellite derived data. Considering MOD16 limitations on mountainous areas, we focus on three low altitude catch-

ments where it may be considered as realistic. Figure 8 shows ET regimes on the available 2000-2005 period. Introduction of a crop coefficient based formulation (V1 and SD versions) has a great impact on ET regimes. Compared to MORDOR V0 reference, ET is increased during spring and summer but decreased in autumn during the end of the growing season. Comparison with MOD16 data suggests that this new seasonality is more realistic, as illustrated by NSE values (see legends of figure 8). In particular it removes the unrealistic increase of ET in autumn during senescence of the vegetation. In this case, spatial

discretisation (MORDOR SD) has a second order impact.



## 5    Conclusions

In this study we validated improvements in an operational hydrological model, using a multi-catchments, multi-criteria and multi-variables framework. From the historical version of the model, two alternative structures have been evaluated. Within the first, physical equations have been revisited to better represent main hydrological components, such as evapotranspiration or

snow, and to reduce model parameters. The second alternative structure integrates this new formulation in an elevation zones spatialization (semi-distributed scheme).

A first evaluation focused on runoff simulation with a multi-criteria split-sample test. Five criteria were identified to focus on various streamflow signatures. For each criterion, the two alternative models perform significantly better than the initial one. On pluvial catchments, improvements are mainly due to the new physical formulation. On the contrary, orographic dis-

cretization provides the main gains on nival catchments. A the end, the new semi-distributed model shows significantly higher performances for runoff simulation for all catchments and for all criteria.

Second evaluation was performed on two independent hydrological variables, not used for model training: snow and evapotranspiration. The objective was to enforce our conclusions, by performing a discharge-independent validation. Results clearly demonstrate model improvement. This semi-distributed structure simulates snow processes in a pretty realistic way. Simulation

of snow cover and snow water equivalent are significantly improved. Realism of the water-balance is also improved in the new model formulation. When compared with satellite proxy, the evapotranspiration dynamic is shown to be deeply improved.

This paper has therefore shown that MORDOR SD provides a very efficient tool for wide-ranging hydrological applications to hydrological simulation in pluvial and nival driven catchments. Performances and versatility of this new model version is very significantly improved. In the same time, its structure has been simplified, specially concerning snow processes, since

the number of free parameters was reduced. Currently, further experience with MORDOR SD is gained as it is implemented in the EDF flood-forecasting chain and in hydrological studies. An assimilation scheme is also being implemented, which integrates both discharge and snow measurement. Future work will focus upon implementation of a fully distributed version of MORDOR SD model over large-scale catchments and in ungauged contexts.

## Appendix A:  MORDOR SD

This section describes in details the MORDOR SD model structure. Figure 9 shows the wiring diagram of MORDOR SD model. It is important to underline that MORDOR V1 equations are exactly the same of MORDOR SD with the only difference that the watershed is not descritized into elevation zones.

### A1    Watershed description

MORDOR SD model is based on a succinct description of the catchment, through the following characteristics : (i) $sbv$ the

watershed area [km²], (ii) $f_{ice}$ relative area of ice [%]; (iii) $f_{lake}$ relative area of lake [%], (iv) $xlat$ latitude of watershed centroid [°], (v) $\bar{fl}$ the mean of flowlength of each gridcell to the outlet [km] and (vi) the average elevation of watershed $\bar{z}$.





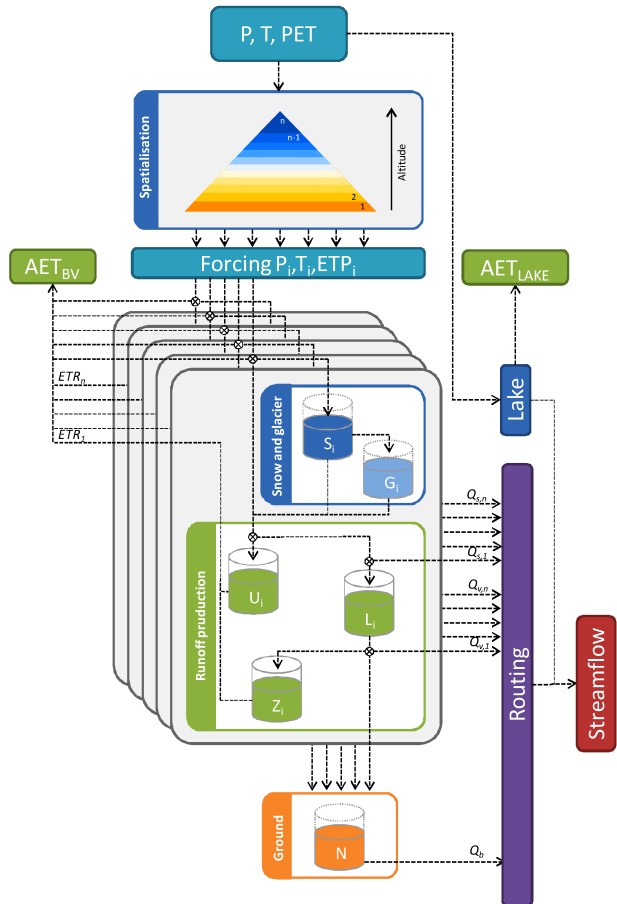

**Figure 9.** Overview of MORDOR SD model components and fluxes.

Furthermore the watershed is descretized into several elevation zones. Each zone $i$ is described by its relative area $s_i$ [%] and its median elevation $z_i$ [m]. Implicitly $\sum_{i=1}^{i=N_b} s_i = 1$, where $i$ is the zone index and $n_b$ is total number of zones. $n_b$ is equal to 1 in the case of MORDOR V1.

## A2   Forcing

5   The model has as input data, for each elevation zone $i$ and timestep $t$, three forcing : (i) precipitation $P_i(t)$; (ii) air temperature $T_i(t)$ and (iii) potential evapotranspiration $PET_i(t)$. Often in the operational context only the areal precipitation $P(t)$ and temperature $T(t)$ are available. In this case the forcing data for each zone are computed through two orographic gradients :

$$P_i(t) = P(t) \cdot (1 + \frac{gpz}{1000}) \cdot (z_i - \bar{z}) \tag{A1}$$

$$T_i(t) = T(t) \cdot \frac{gtz}{100} \cdot (z_i - \bar{z}) \tag{A2}$$





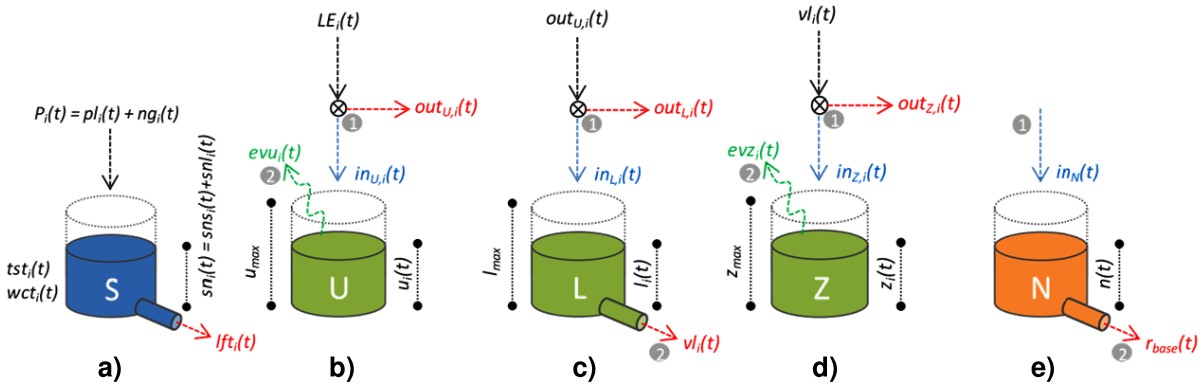

**Figure 10.** Schematic representation of MORDOR SD storage.

where $gpz$ is the precipitation gradient [%/1000m] and $gtz$ is the temperature gradient [°/100m]. In this case the $PET_i(t)$ could be computed with several formula driven by $T_i(t)$, for instance following the formula proposed by Oudin et al. (2005). These equations are not used in MORDOR V1.

**A3 Water balance**

From the reference potential evapotranspiration $ET0_i(t)$, a Maximal Evapotranspiration $MET_i(t)$ is computed using a crop coefficient $Kc$, such as:

$$MET_i(t) = Kc \cdot ET0_i(t) \tag{A3}$$

In its classical form, the $Kc$ coefficient is defined for any crop using look-up tables (Allen et al., 1998). However, in an operational and meso-scale context, a watershed effective $Kc$ must be defined, in order to accommodate with various hydrological contexts and to efficiently supply the water-balance. In the model, $Kc$ formulation is:

$$K_c^i = K_{min} + (1 - K_{min}) \cdot \frac{(Rpot_i - \min(Rpot))}{(\max(Rpot) - \min(Rpot))} \tag{A4}$$

$$K_c^i = \frac{K_c^i}{\overline{K_c^i}} \tag{A5}$$

With $K_{min}$ the minimal seasonal crop coefficient value and $Rpot$ the potential solar radiation. From the $MET$, the model calculates the actual evapotranspiration ($AET$) from three components: (i) surface interception $ev0_i(t)$ according to the following formula : $ev0_i(t) = min(MET_i(t), P_i(t))$; (ii) evapotranspiration from the root soil $evu_i(t)$, see A5.1; (iii) evapotranspiration from the capillarity water storage in the hillslope, see A5.3




## A4  Snow module

The aim of storage $S$ is to model the snow pack. The figure 10a show the I/O and the state variables of this storage.

### A4.1  Snow accumulation

For each elevation zone $i$ and timestep $t$, the precipitation $P_i(t)$ is divided in two components : (i) the liquid part $pl_i(t)$, i.e. rain, and (ii) the solid part $ng_i(t)$, i.e. snow. Then the input of the storage $S$ are:

$$pl_i(t) = fliq(t) \cdot P_i(t) \tag{A6}$$
$$ng_i(t) = (1 - fliq(t)) \cdot P_i(t) \tag{A7}$$

where $fliq(t)$ is the liquid ratio of precipitation founded on classical parametric S-shaped curve:

$$fliq_i(t) = 1 - [1 + exp(\frac{10}{\delta T} \cdot ((T_i(t) + efp) - t_{50}))]^{-1}] \tag{A8}$$

where $\delta T$ is the thermic range (set to 4 °C), $t_{50}$ is the threshold temperature between solid and liquid phases (set to 1 °C) and $efp$ is an additive correction parameter, by default set equal to zero.

### A4.2  Snow melt

For each elevation zone $i$ and each timestep $t$, the snow pack is summarised by two state variables : the bulk temperature $tst_i(t)$ and the water content $wct_i(t)$. The snow pack temperature is computed using an exponential smoothing function as follow:

$$tst_i(t) = \min\{lts \cdot tst_i(t-1) + (1 - lts) \cdot (T_i(t) + efp), 0\} \tag{A9}$$

where $lts$ is the smoothing parameter between the antecedent snow pack temperature and the actual modified air temperature. The melt runoff $lft_i(t)$ is composed by two parts : the surface melt $lft_i(t)$ and the ground melt $gm$. This last considered as constant in time and space. The surface melt change according the elevation zone $i$ and the timestep $t$ as follow:

$$lfs_i(t) = K_f \cdot (T_i(t) + eft + tst_i(t)) \tag{A10}$$

where $K_f$ is the melting coefficient and $eft$ is additive correction parameter, by default set to zero. The melting coefficient is computed via this equation :

$$K_f = k_f + (k_{fp} \cdot \frac{R_{pot}(t)}{\overline{R_{pot}}}) \tag{A11}$$

where $k_f$ is the fixed part and the $k_{fp}$ the variable part of melting coefficient. For a given elevation zone $i$ and the timestep $t$, the output of the snow model is the runoff $le_i(t)$ equal to the sum of the rainfall $p_i(t)$, the surface melt $lfs_i(t)$ and the ground melt $gm$.



## A5 Runoff production

### A5.1 Surface storage U

The storage $U$ is intended to represent the water absorption capacity of root zone. As shown in figure 10b, the I/O of storage $U$ follow these equations:

$$in_{U,i}(t) = (u_{max} - u_i(t-1)) \cdot (1 - \exp(-\frac{le_i(t)}{u_{max}})) \tag{A12}$$

$$out_{U,i} = le_i(t) - in_{U,i}(t) \tag{A13}$$

$$ev_{U,i}(t) = (MET_i(t) - ev0_i(t)) \cdot \frac{u_i(t)}{u_{max}} \tag{A14}$$

where $u_i(t)$ is the water content of storage $U$ for the elevation zone $i$ at the timestep $t$ and $u_{max}$ is the maximum capacity of the storage, assumed constant for all the zones. This parameter is assumed to be the same for all the zones.

### A5.2 Hillslope storage L

The storage $L$ is intended to represent the hillslope zone. As shown in figure 10c, the I/O of storage $L$ follow these equations:

$$in_{L,i}(t) = out_{U,i}(t) \cdot [1 - (\frac{l_i(t-1)}{l_{max}})^2] \tag{A15}$$

$$out_{L,i} = out_{U,i}(t) - in_{L,i}(t) \tag{A16}$$

$$v_{L,i}(t) = k_L \cdot l_i(t)^{evl} = \frac{1}{evl \cdot l_{max}^{evl-1}} \cdot l_i(t)^{evl} \tag{A17}$$

where $l_i(t)$ is the water content of storage $L$ for the elevation zone $i$ at the timestep $t$ and $L_{max}$ is the maximum capacity of the storage, assumed constant for all the zones. The parameter $evl$ is the outflow exponent. Then, the surface runoff, $r_{surf,i}(t)$, provided by the elevation zone $i$ is computed according to:

$$r_{surf,i}(t) = out_{L,i}(t) - max(0, in_{L,i}(t) - l_{max}) \tag{A18}$$

### A5.3 Capillarity storage Z

The storage $Z$ is intended to represent the capillarity of the hillslope zone. As shown in figure 10d, the I/O of storage $Z$ follow these equations :

$$in_{Z,i}(t) = v_{L,i}(t) \cdot [1 - (\frac{z_i(t-1)}{z_{max}})] \tag{A19}$$

$$out_{Z,i} = v_{L,i}(t) - in_{Z,i}(t) \tag{A20}$$

$$ev_{Z,i} = (MET_i(t) - ev0_i(t) - ev_{U,i}(t)) \cdot \frac{z_i(t)}{z_{max}} \tag{A21}$$

where $z_i(t)$ is the water content of storage $Z$ for the elevation zone $i$ at the timestep $t$ and $Z_{max}$ is the maximum capacity of the storage, assumed constant for all the zones. Then, the sub-surface runoff, $r_{vers,i}(t)$, provided by the elevation zone $i$ is



computed according to:

$$r_{vers,i}(t) = k_r \cdot out_{Z,i}(t) \tag{A22}$$

where $k_r$ is the runoff coefficient, ranging from 0 to 1.

### A5.4 Ground storage N

The deep storage $N$ determines the baseflow runoff. As shown in figure 10e, the I/O of storage $N$ follow these equations :

$$in_N(t) = \sum_{i=1}^{N_b} ((1 - k_r) \cdot out_{Z,i}(t)) \cdot s_i \tag{A23}$$

$$r_{base}(t) = k_N \cdot n(t)^{evn} \tag{A24}$$

### A6   Routing function

The model identifies three flux components: (i) surface runoff $r_{surf}$; (ii) sub-surface exfiltration $r_{vers}$; (iii) base flow $r_{base}$.
The global streamflow $rt(t)$ is the sum of these three components, as follow:

$$rt(t) = (\sum_{i=1}^{N_b} r_{surf,i}(t)) \cdot s_i + (\sum_{i=1}^{N_b} r_{vers,i}(t)) \cdot s_i + r_{base}(t) \tag{A25}$$

The routing function used to transfer the global streamflow to the outlet is based on the diffusive wave equation (Hayami, 1951):

$$f(t, cel, dif) = \frac{\bar{fl}}{2\sqrt{\pi dif}} \cdot t^{-\frac{3}{2}} \cdot e^{-\frac{(fl - cel \cdot t)^2}{4 \cdot dif \cdot t}} \tag{A26}$$

where $t$ is the timestep, $cel$ is the celerity of the wave in $km/h$ and $dif$ is the diffusion of the wave in $km^2/h$.



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
