# Peer review of "Impact of model structure on flow simulation and hydrological realism: from lumped to semi-distributed approach."

_Hydrology and Earth System Sciences, 2017_

## Referee Comment (RC1) · Anonymous Referee #1 · 23 Feb 2017

The manuscript "Impact of model structure on flow simulations and hydrological realism: from lumped to semi-distributed approach" reports on the performance of three different versions of the MORDOR hydrological model. The model versions consist of the initial lumped model MORDOR v0 developed in the 1990s, the revised lumped version v1, which includes new formulations for calculating evapotranspiration and a more complex snow model, and finally the version MORDOR SD, which is basically a semi-distributed version of MORDOR v1. The evaluation of the model performances is based on a multi-criteria split-sample test, utilizing a comprehensive data set from 50 different catchments in France. For the evaluation of runoff, 5 different streamflow signatures are used, of which three are applied in the objective function for model calibration. Additionally, an independent evaluation is performed with MODIS snow cover data, measured snow water equivalent time series and actual evapotranspiration data of the MOD16 AET product. The results show noteworthy improvements of the simulation performances for the versions v1 and MORDOR SD, especially when evaluation the streamflow signatures. MORDOR SD logically also outperforms the lumped models when it comes the simulation of snow cover and snow water equivalent. Although the new model versions show improved representation of actual evapotranspiration, no clear advantages are found for the semi-distributed version of the model. Surprisingly, the simulation results showed good agreement with the data sets not used for model training.

I enjoyed reading the manuscript. It is straightforward, logically structured and mostly well written. The presented models are not necessarily revolutionary but are used in operational hydrological studies and flood-forecasting activities in EDF (French electric utility company), which belongs to the three largest energy producers on the globe. It is of interest to read about model development activities outside the classic academic circle, having in mind that the applied models have clear implications on society and decisions made in practice. The topics and findings are therefore of interest for the (scientific) community and a publication is sincerely recommended.

There are however some points which need to be addressed and where additional clarifications and analysis would be useful and are needed.

**General comments:**

1.  The authors show interesting and relevant simulation results that are in good agreement with data sets not used in calibration (FSC, SWE, ETA). These detailed results are however only presented for 6 out of 50 catchments. It is clear that (i) snow is not relevant in all catchment, (ii) availability of SWE measurements may be limited and (iii) that the ETA estimates from MOD16 are unreliable in alpine regions. It would however be of interest to see results of the model performance for more catchments in this context.

2. The paper reports on different versions of the hydrological model MORDOR. It would therefore be important for more clarity to add two overview tables, containing (i) model parameters, units, range of parameter values and description and (ii) model fluxes and states, also including units and a description of the variables.

3. MORDOR v1 and SD include a modified and improved snow routine. How is snow sublimation considered in the models?

4. From the manuscript it is unclear, what temperature data is used.

5. How is PET / ET0 estimated? How large are the differences between the PET values in version v0 compared to version v1/SD? This is crucial since it will influence the AET results.

6. What is the reason to use KGE in the objective function and NSE for evaluation?

7. The authors state that the model runs at different temporal resolution. Are the calibrated parameters comparable between the different temporal resolutions?

8. In the Appendix the model formulations are given. Here it would also be interesting to give some technical details on the models: In what language are they written? Is there an internal time discretisation implemented? How long does a run take?

**Specific comments:**

P1L22: semi-distributed

P2L1: Most studies

P2L18-19: It is unclear why the two best solutions are selected (I presume the presented MORDOR v1 and SD) and why these three new formulations are then compared with the historical version.

P2L23: …in quality and length of available records.

Table 1: The number of free parameters are not underlined in the table.

Table 1: Could you please clarify what is to be understood under "adjusted PET from a statistical formulation driven by temperature"?

Figure 1: It would be good to highlight and name the catchments shown in detail in the analysis.

P5L23: Why not call the section 3.1.2 simply MORDOR V1

P5L24: revised model formulation

P6L18: … time-step based on air temperature.

Figure 2: A subdivision in 10 elevation zones is shown. Is this the standard number of zones used for spatial discretisation, also for the other catchments?

P8L7: The term "streamflow interannual daily regime" is not easily understandable.

P8L8: Do you mean the flow duration curve with "streamflow empirical cumulative distribution"?

P8L9: How are the streamflow recession periods defined in practice?

P8L10: What is meant with the $1^{st}$-lag streamflow derivates and how are they calculated?

P8L11: … performance results are resumed using… What is meant with resumed?

P8L27: (i.e. 100 simulations)

P9L8: in figure 4

Figure 4: Legend is missing

Figure 7: Why is $R^2$ shown and not NSE (as in other plots)?

P12L7: (see Figure 2?)

P12L20: What is the second order impact? Not being important?

*Appendix A*

Additional tables as mentioned above showing parameters, fluxes and states would very much improve clarity when reading the appendix. It is very frequently unclear what the used variables mean. Figure 10 does not help very much in this context. Following are a few things a noticed, being however sure, that the list is probably incomplete:

P13L31: What is meant with "flow length of each gridcell to the outlet", since we are talking about zones?

Eq. A1 & A2: Please check equations. They seem to be erroneous.

Eq. A3: Why ET0 and not PET? This is not consistent with the other parts of the manuscript.

Eq. A4: Should $K_c$ and $R_{pot}$ not have a time component?

Eq. A5: It is unclear, for what A5 is needed.

Eq. A11: $K_f$: Is the time component missing? It is unclear, what the difference between $k_f$ and $k_{fp}$ is.

Eq. A14: From the equations it is unclear, how $u_i(t)$ is calculated.

Eq. A21: From the equations it is unclear, how $z_i(t)$ is calculated.

Eq. 23/25: What is $s_i$?

Eq. 24: What is $k_N$?

Eq. 26: What is fl?

M. Herrnegger, 22.02.2017

---

## Referee Comment (RC2) · M. Hrachowitz (Referee) · 29 Mar 2017

The manuscript "Impact of model structure on flow simulation and hydrological realism: from lumped to semi-distributed approach" by Garavaglia et al., analyses the respective values of three different model formulations (2 lumped, 1 semi-distributed) for the prediction of snow dynamics and seasonal evaporation pattern.

I really liked the approach and enjoyed reading the manuscript, which is well-structured and, as far as I can tell, relying on a sound methodology. The overall topic is of high relevance, as it directly addresses the urgent need of the modelling community to improve their predictions (and thus their models), and as it picks up a much advocated and potentially fruitful way forward: the use of semi-distributed formulations together

with more efficient use of available data within multi-objective/multi-variable model calibration and post-calibration evaluation techniques. I would therefore be glad to see this work eventually published, but I would nevertheless strongly encourage the authors to address a couple of points which will strengthen the manuscript.

(1) The manuscript will benefit from being proof-read by a native English speaker to reduce the number of typos and language errors (grammar, syntax and use of specific words/terms)

(2) It will be of tremendous help for the reader if the author provided tables of

(a) the catchments used (including names, geographical positions, catchment areas, elevation range, slopes, annual P, annual potential E, annual Q, modelling time period, and time step

(b) the parameters of each model, the associated symbols, units, prior distributions (are these the same for all catchments?) and descriptions

(c) all model components (i.e. states and fluxes), including their symbols, dimensions and descriptions. This would make it much more convenient to follow the Appendix, in which many symbols are not clearly defined at this point.

If deemed suitable, these tables can be provided as Supplementary Material.

(3) Section 3.2.2 will benefit from a clearer description of the different criteria. For example, it remains unclear what is meant by "streamflow regime". I suppose it is the long-term seasonal pattern, but please make this more specific. Similarly, the cumulative distribution of flows is commonly referred to as flow-duration curve. A more consistent terminology will help the reader to better appreciate the manuscript. It is also not clear what is meant by 1st-lag flow derivative. Does this refer to the lag-1 autocorrelation? Of flows? Of the recession? Please elaborate!

(4) The post-calibration evaluation of the models with respect to snow and evaporation dynamics is an important point in this paper. Yet, no mention of this is made in section

3.2.2. How are MODIS data used to compare to model output? Spatial averages? What about the temporal resolution of the evaluation? Which performance metric was used? Some of this is mentioned later in the manuscript but I think this needs to be made clear in the methods section.

(5) Related to (4), I did not understand how a fractional snow cover can be reproduced with lumped model formulations (VO and V1). This makes clearly sense for a semi-distributed model (Mordor SD). But obviously I missed something for the lumped versions. Please clarify!

(6) What is the reason behind using KGE for calibration (which is completely fine) but NSE for evaluation? Why is not the same metric used for both?

(7) The presentation of the results and discussion section would strongly benefit from a bit more detail. Detailed results are only shown for a few catchments with good overall performance. And even for these, it remains unclear how the modelled hydrograph looks like (in comparison to the observed one) and what the values of the individual associated calibration objective functions (i.e. the 3 individual KGEs) and evaluation metrics (the remaining criteria) are. In addition, I think it would also be valuable to show examples of catchments where the model adaptation did not work and also discuss why.

(8) Related to (7), it is mentioned that V1 provides substantial improvements compared to V0. As V1 is changed in various respects in comparison to V0, it would be great if the authors invested a bit of effort to analyse and document which part/adjustment of V1 contributes most to the improvement.

(9) P.1,l.6: what is meant by "inflected"? Please rephrase.

(10) P.1,l.8: should read as "...evapotranspiration estimates. The model comparison is...."

(11) P.1,l.22: should read as "...semi-distributed..."

(12) P.1,l.23: Nijzink et al. (2016) would fit in nicely here

(13) P.1,l.23: what is meant by "To overpass hydrological singularity...."? Please rephrase.

(14) P.2,l.8: I may be worth referring to Hrachowitz et al. (2014) here.

(15) P.2,l.15: should read as "...framework on the MORDOR...."

(16) P.2,l.20-22: irrelevant. Can be condensed.

(17) P.2,l.26: should read as "......mainly in the Alps (18 catchments), the Pyrenees (5 catchments) and the Massif Central..."

(18) P.2,l.29: should read as "...hydrological conditions. The average area of the study catchments is..."

(19) P.3,Table 1: not clear if the 22/17/19 parameters are all calibration parameters, as it seems in the Appendix that some of them are fixed. Please clarify.

(20) P.3,l.4: should read as "...1635 mm/yr. With regard to...."

(21) P.3,l.10: should read as "...sub-daily time steps..."

(22) P.3,l.11: what is meant by " the shape of local gauges"? Please clarify.

(23) P.3,l.12ff: that is ok, but it should be underlined that these are not observations but modelled estimates which can be subject to considerable uncertainty.

(24) P.4,l.1-2: should read as "...for being affected by many..."

(25) P.4,l.5: should read as "...provides fractional snow cover..."

(26) P.4,l.5: please explain what "fractional snow cover" describes. Are these spatial fractions? If yes across the entire catchment? Across a pixel? Which value was used to compare the modelled values with?

(27) P.4,l.15: should read as "...interconnected storages."

(28) P.4,l.15: what is meant by "continuously"? Please clarify.

(29) P.5,l.1: No, what is required is a *representative* estimate of areal precipitation. The mean (or any other measure of central tendency) will average out extremes, which will, due to the non-linear nature of your (or better: any meaningful hydrological model), result in biased results.

(30) P.5,l.19: "(ii) snow modelling have to be improved..." reads awkward. Please rephrase.

(31) P.5,l.31: should read as "...evapotranspiration, the model..."

(32) P.5,l.32: what is meant by "neutralized"??

(33) P.6,l.1: It is not clear which part of the system the ground-melt component represents. What exactly does it do? Please clarify.

(34) P.6,l.30: that is fine, but please specify if the gradients are set to fixed values or if they are calibrated (similar to rainfall multipliers). Where do the values (fixed or prior distributions) come from? Literature? Please provide references.

(35) P.7,l.2,section 3.2: I would suggest to rearrange this section for a better flow and to start with the calibration approach, followed by the split sample test and the post-evaluation criteria.

(36) P.7,l.5: does this mean that you end up with 2 parameter sets for each catchments? Is the following analysis then based on these 100 parameter sets (i.e. 2 for each catchment)? Please describe in more detail what you are doing.

(37) P.8,l.1-2: this resembles an approach described by Gharari et al. (2013). It would be good to refer to that paper.

(38) P.8,l.4ff: please clearly separate between criteria that are used for calibration (i.e. q, reg and qlc) and those used for post-calibration evaluation (i.e. etg, dq, snow cover, evaporation).

(39) P.8,l.17, eq.1: should this not read "KGEqcl"?

(40) P.8,l.20: "Numerous applications if this OF…" please provide references.

(41) P.8,l.29: Do the V1 and SD models in \*all\* catchments outperform V0 or is it just on average? Please provide some representative examples for both – cases of improvements and cases where V1 and SD did not result in improvements

(42) P.9,l.4: the improvement is obvious, but I struggle to see the "spectacular" improvement. In addition, "most" seems also a bit exaggerated here: reg, qcl and etg show only minor improvements, if any. Please tone the statement down a bit to actually reflect what we can see in the figures.

(43) P.10, Figure 5: are the NSE values the NSE values of the snow cover? Please clarify. In addition, please make sure that \*all\* figure captions in the manuscript are stand-alone, i.e. that the reader can fully understand a figure only by reading its caption.

(44) P.10,l.1: what is meant by "overpasses"? please rephrase.

(45) P.10,l.2: what is meant by "…the interest of the…"? please rephrase.

(46) P.11,figures 6,7: see (43)

(47) P.12,l.2: should read as "…that cannot be…"

Best regards, Markus Hrachowitz

References:

Gharari, S., Hrachowitz, M., Fenicia, F., & Savenije, H.H.G. (2013). An approach to identify time consistent model parameters: sub-period calibration. Hydrology and Earth System Sciences, 17(1), 149-161.

Hrachowitz, M., Fovet, O., Ruiz, L., Euser, T., Gharari, S., Nijzink, R., Freer, J., Savenije, H.H.G. & Gascuel‐Odoux, C. (2014). Process consistency in models:

[Figure]

The importance of system signatures, expert knowledge, and process complexity. Water Resources Research 50(9): 7445-7469.

Nijzink, R.C., Samaniego, L., Mai, J., Kumar, R., Thober, S., Zink, M., Schäfer, D., Savenije, H.H.G. & Hrachowitz M. (2016). The importance of topography-controlled sub-grid process heterogeneity and semi-quantitative prior constraints in distributed hydrological models. Hydrology and Earth System Sciences, 20(3), 1151-1176.

---

## Author Comment (AC1) · 16 May 2017

**Detailed response to the comments of referee 1**

We want to thank referee 1 (M. Herrnegger) for his accurate and helpful review of our manuscript. In this author comment, we list how each of the remarks provided by the referee was addressed. The comments made by the referee will be referred as RC and printed in bold; the authors' comments and answers as AC.

**Concerning general comments:**

**1 RC: The authors show interesting and relevant simulation results that are in good agreement with data sets not used in calibration (FSC, SWE, ETA). These detailed results are however only presented for 6 out of 50 catchments. It is clear that (i) snow is not relevant in all catchment, (ii) availability of SWE measurements may be limited and (iii) that the ETA estimates from MOD16 are unreliable in alpine regions. It would however be of interest to see results of the model performance for more catchments in this context.**

AC: We agree. We propose to extend the results to 8 catchments for FSC (a compromise between nival behaviour and data availability). For AET, we also propose to extend the results to 8 catchments (a compromise between pluvial behaviour and data availability). Unfortunately for SWE, we are not able to present other results due to data availability of at site snow gauges.

**2 RC: The paper reports on different versions of the hydrological model MOR-DOR. It would therefore be important for more clarity to add two overview tables, containing (i) model parameters, units, range of parameter values and description and (ii) model fluxes and states, also including units and a description of the variables.**

AC: To address this issue we added a supplementary table in Appendix A which summarize MORDOR V1/SD free parameters, units, prior range and description. In addition we completed the description of model fluxes and states in Appendix A. Table 1 was also improved. On the other hand, concerning historical model version (MORDOR V0) we only added explicit references to existing publications which describe the model.

**3 RC: MORDOR v1 and SD include a modified and improved snow routine. How is snow sublimation considered in the models?**

AC: Snow sublimation is not taken into account in the models. Although it can be a significant process at local scale, it is considered to be of second-order at catchment scale in our regions. See for example Strasser et al. (2008).

**4 RC: From the manuscript it is unclear, what temperature data is used**
AC: The temperature data used in the study are gridded and provided by (Gottardi et al. 2012). These data result from a statistical reanalysis based on ground network data and weather patterns (Garavaglia et al. 2010). They are available for the 1948-2012 period at 1-km$^2$ / 1-day resolution. See P3L3-6.

**5 RC: How is PET / ET0 estimated? How large are the differences between the PET values in version v0 compared to version v1/SD? This is crucial since it will influence the AET results.**
AC: PET is estimated differently for the three model structures.
For V0, PET is calculated from a statistical formulation driven by air temperature $T$, as follows:
$$PET = a.(T - b)^2$$
with a and b two free parameters which are calibrated with the other model parameters.
For V1/SD, PET may be estimated by any PET formula. In this study, we use the Oudin formulation, which is expressed as follows:
$$PET = 0.408.Rpot.\frac{T + 5}{100}$$
with $Rpot$ the potential solar radiation ($MJ.m^{-2}.d^{-1}$) and T the air temperature ($C$).
On the study catchments, Oudin PET vary from about $420\ mm.yr^{-1}$ to $890\ mm.yr^{-1}$.
On the other hand, MORDOR V0 calibrated PET vary from about $220\ mm.yr^{-1}$ to $1750\ mm.yr^{-1}$.
We agree with the reviewer, these differences in the PET estimates obviously have a

great impact on AET results. We added a specific comment in section 4.3.

**6 RC: What is the reason to use KGE in the objective function and NSE for evaluation?**
AC: We use the KGE for calibration because of its good statistical properties, which are helpfull for parameters identification. On the other hand, model evaluation is based on NSE because this criterion is commonly used for evaluation of hydrological models and is therefore suitable to use as a benchmark for this study. In addition, it allows to consider different metrics for calibration and posterior evaluation.

**7 RC: The authors state that the model runs at different temporal resolution. Are the calibrated parameters comparable between the different temporal resolutions?**
AC: Most parameters do not depend on temporal model resolution. However some parameters like the unit hydrograph parameters and the parameters used in the $L$ storage outflow equation remain dependent on the temporal resolution. Concerning the model calibration process, we use a wide prior range for parameters values which is relevant for both daily and sub-daily time steps.

**8 RC: In the Appendix the model formulations are given. Here it would also be interesting to give some technical details on the models: In what language are they written? Is there an internal time discretisation implemented? How long does a run take?**
AC: We propose to add this paragraph in the Appendix : "The MORDOR SD model is written in FORTRAN 90. The model runs at different temporal resolution. The duration of a simple model simulation (i.e. model run and evaluation criteria computation) is about 1 sec and depends on the time step and on the length of time series. For instance a daily simulation over 50 years takes less than 1 sec and an hourly

simulation over 10 years takes about 2 sec. Concerning the calibration process (about 40.000 model runs), the algorithm takes about 10 min for a daily time step over 50 years and about 45 min for an hourly time step over 10 years. Post-processing and graphical tools are developed in R language."

**Concerning specific comments:**

**RC: P1L22: semi‐distributed**
AC: It has been changed in the revised manuscript.

**RC: P2L1: Most studies**
AC: It has been changed in the revised manuscript.

**RC: P2L18‐19: It is unclear why the two best solutions are selected (I presume the presented MORDOR v1 and SD) and why these three new formulations are then compared with the historical version.**
AC: "Three" has been replaced with "two" in the revised manuscript.

**RC: P2L23: . . .in quality and length of available records.**
AC: It has been changed in the revised manuscript.

**RC: The number of free parameters are not underlined in the table.**
AC: We propose a new redaction of the legend of Table 1 : ". . . For each module and model, the number of free parameters is given."

**RC: Table 1: Could you please clarify what is to be understood under "adjusted**

PET from a statistical formulation driven by temperature"?
AC: See response to RC 5.

**RC: Figure 1: It would be good to highlight and name the catchments shown in detail in the analysis.**
AC: We propose to provide as Supplementary Material : (i) a table of the main features of the catchments used, including name, geographical position (i.e. coordinates of the outlet), catchment area, elevation range, slope, annual P, annual PET, annual Q, time step, modelling periods P1 and P2; (ii) a specific figure with the catchments location.

**RC: P5L23: Why not call the section 3.1.2 simply MORDOR V1**
AC: We propose to homogenize the three sections titles as follows:
3.1.1. MORDOR V0: Initial lumped formulation
3.1.2. MORDOR V1: Revised lumped formulation
3.1.3. MORDOR SD: Semi-distributed formulation

**RC: P5L24: revised model formulation**
AC: It has been changed in the revised manuscript.

**RC: P6L18:** … time‐step based on air temperature.
AC: It has been changed in the revised manuscript.

**RC: Figure 2: A subdivision in 10 elevation zones is shown. Is this the standard number of zones used for spatial discretisation, also for the other catchments?**
AC: No. In this study, the number of elevation zones depends on the hypsometric curve of the catchment according to the following criteria: (i) the relative area of each elevation zone has to be greater or equal to 5% and less or equal to 50%, (ii) the

elevation range of each zone has to be lower than 350m.
We propose to add this comment in section 3.1.3.

**RC: P8L7: The term "streamflow interannual daily regime" is not easily understandable.**
AC: We agree with your comment, so we propose a new formulation for the section 3.2.2, as follows:

"The runoff signatures are viewed in such a way that streamflow data may be broken up into several samples, each of them a manifestation of catchment functioning. Five different signatures are used in this study and are described in the following:

- time serie of flow is obviously the first signature which has to be reproduced by the model (hereafter called $Q$);

- long-term mean daily streamflow is used to focus on the capacity to reproduce seasonal variation of observations (hereafter called $Qsea$);

- flow duration curve focuses on the capacity to reproduce streamflow variance and extremes (hereafter called $FDC$);

- flow recessions during low flow period focuses on streamflow recessions (hereafter called $Qlow$);

- $lag - 1$ streamflow variation is the last signature focusing on short term variability (hereafter called $dQ$ and computed as follows: $dQ(t) = Q(t) - Q(t-1)$).

**RC: P8L8: Do you mean the flow duration curve with "streamflow empirical cumulative distribution"?**

AC: Yes. See comment above.

**RC: P8L9: How are the streamflow recession periods defined in practice?**
AC: Streamflow recessions sequences are extracted from the low flow period for each catchment. We only select recessions with a minimum duration of 7 days.

**RC: P8L10: What is meant with the 1 st ‐lag streamflow derivates and how are they calculated?**
AC: Yes. See comment above.

**RC: P8L11: ... performance results are resumed using... What is meant with resumed?**
AC: We replace "resumed" by "summarized"

**RC: P8L27: (i.e. 100 simulations)**
AC: It has been changed in the revised manuscript.

**RC: P9L8: in figure 4**
AC: It has been changed in the revised manuscript.

**RC: Figure 4: Legend is missing**
AC: We added the legend in figure 4.

**RC: Figure 7: Why is $R^2$ shown and not NSE (as in other plots)?**
AC: Your remark is relevant. We therefore change the figure criteria, with NSE instead of $R^2$. The conclusion remains the same.

**P12L7: (see Figure 2?)**
AC: It has been changed in the revised manuscript.

**RC: P12L20: What is the second order impact? Not being important?**
AC: We replaced by "... has a second order effect".

**Concerning Appendix A:**

**RC: Additional tables as mentioned above showing parameters, fluxes and states would very much improve clarity when reading the appendix. It is very frequently unclear what the used variables mean. Figure 10 does not help very much in this context.**
AC: See response to comment RC 2.

**RC: P13L31: What is meant with "flow length of each gridcell to the outlet", since we are talking about zones?**
AC: The $\bar{fl}$ parameter is the average of the flow length of each DEM pixel to the outlet. This parameter is used in the routing function (diffusive wave equation, see A.6) that remains lumped, i.e. there is no relation between $\bar{fl}$ and the elevation zones.

**RC: Eq. A1 & A2: Please check equations. They seem to be erroneous.**
AC: We checked and we corrected equation A2.

**RC: Eq. A3: Why ET0 and not PET? This is not consistent with the other parts of the manuscript.**

AC: We agree. We propose to replace ET0, which has a very precise acceptation, by PET.

**RC: Eq. A4: Should Kc and Rpot not have a time component?**
AC: We agree. We changed equation A4 and A5.

**RC: Eq. A5: It is unclear, for what A5 is needed.**
AC: A5 equation is needed to have a mean value of $K_C$ equal to 1, whatever the $kmin$ parameter value. In fact, $kmin$ is a shape parameter representing the seasonal variability of the vegetation, and we want to reduce the sensitivity of the evapotranspiration amount to $kmin$.

**RC: Eq. A11: Kf : Is the time component missing? It is unclear, what the difference between kf and kfp is.**
AC: $K_f$ is a time variable coefficient. We therefore modified equations A10 and A11. We also added a specific comment to clarify $k_f$ and $kf_p$ terms.

**RC: Eq. A14: From the equations it is unclear, how ui(t) is calculated.**
AC: As illustrated on Figure A2, $u_i(t)$ is calculated as follows :

$$u_i(t) = u_i(t-1) + in_{U,i}(t) - evu_i(t)$$

**RC: Eq. A21: From the equations it is unclear, how zi(t) is calculated.**
AC: As illustrated on Figure A2, $z_i(t)$ is calculated as follows :

$$z_i(t) = z_i(t-1) + in_{z,i}(t) - evz_i(t)$$

**RC: Eq. 23/25: What is si ?**

AC: $s_i$ is the relative area of the zone $i$, see P14L1

**RC: Eq. 24: What is $k_n$ ?**
AC: The parameter $k_n$ is the outflow coefficient of the deep storage $N$. We added a specific comment in Appendix A.

**RC: Eq. 26: What is $fl$?**
AC: $fl$ is the mean of flowlength of each gridcell to the outlet [km], see P13L31 and comment above.

**References**
Strasser, U., Bernhardt, M., Weber, M., Liston, G. E., Mauser, W. (2008). Is snow sublimation important in the alpine water balance?. The Cryosphere, 2(1), 53-66.

**Supplement:**

**S1 Catchments features**

This section details the dataset of the 50 catchments. Figure S1 shows the localization of the studied catchments. The catchments IDs are highlighted in figure S1. Table S1 presents the main features of the catchments dataset, including name, geographical position, area, elevation range, slope, annual P, annual PET, annual Q, time step, modeling periods P1 and P2.

[Figure]

**Figure S1:** Localization of studied catchment, the catchment IDs are highlighted.

**Table S1 :** Main features of the 50 catchments, including name, geographical position, area, elevation range, slope, annual P, annual PET, annual Q, time step, modeling periods P1 anP2.

| ID | BV | Xe [m] | Ye [m] | Surface [km²] | Latitude [degree] | Zmin [m] | Z50 [m] | Zmax [m] | Slope [degree] | Annual_P [mm] | Annual_ETP [mm] | Annual_Q [mm] | Time step [hour] | Modelling Period 1 | Modelling Period 2 | Mountain region |
|---|---|---|---|---|---|---|---|---|---|---|---|---|---|---|---|---|
| 1 | Agout@Fraisse | 637989 | 1845758 | 46.8 | 43.4 | 793 | 999 | 1149 | 5.05 | 1573 | 660 | 1073 | 24 | 1973 - 1981 | 1981 - 1988 | Massif Central |
| 2 | Agout@LaRaviege | 621096 | 1843766 | 370.8 | 43.6 | 659 | 904 | 1259 | 6.37 | 1510 | 657 | 969 | 24 | 1973 - 1980 | 1980 - 1987 | Massif Central |
| 3 | Allier@Poutes | 705537 | 1994942 | 1016.1 | 46.5 | 659 | 1146 | 1528 | 5.41 | 1046 | 604 | 496 | 24 | 1988 - 1996 | 1996 - 2004 | Massif Central |
| 4 | Ardeche@Sauze | 776846 | 1926227 | 2266.5 | 44.1 | 49 | 599 | 1671 | 9.51 | 1574 | 757 | 838 | 24 | 1961 - 1985 | 1985 - 2008 | Massif Central |
| 5 | Arn@Taillades | 619482 | 1836883 | 82.4 | 43.5 | 681 | 880 | 1054 | 4.08 | 1630 | 683 | 1045 | 24 | 1987 - 1997 | 1997 - 2007 | Massif Central |
| 6 | Arve@Arthaz | 904338 | 2135177 | 1638.2 | 46.2 | 433 | 1504 | 4800 | 19.61 | 1552 | 569 | 1438 | 24 | 1986 - 1993 | 1993 - 2000 | Alps |
| 7 | Behine@LaPoutroie | 953246 | 2358462 | 38.4 | 48.2 | 406 | 856 | 1215 | 12.58 | 1316 | 611 | 805 | 24 | 1984 - 1989 | 1989 - 1993 | Vosges |
| 8 | Breze@Meyrueis | 688063 | 1909099 | 33.7 | 44.0 | 703 | 1075 | 1544 | 13.41 | 1444 | 622 | 843 | 24 | 1985 - 1996 | 1996 - 2006 | Massif Central |
| 9 | Bromme@Brommat | 627263 | 1980786 | 106.3 | 44.8 | 660 | 978 | 1610 | 5.70 | 1422 | 696 | 972 | 24 | 1989 - 1995 | 1995 - 2000 | Massif Central |
| 10 | Ceze@Bagnols | 782869 | 1910402 | 1119.2 | 44.1 | 45 | 329 | 1589 | 6.21 | 1330 | 819 | 564 | 24 | 1994 - 1999 | 1999 - 2004 | Massif Central |
| 11 | Chassezac@SteMarguerite | 731606 | 1942481 | 414.1 | 44.3 | 329 | 1055 | 1671 | 11.03 | 1611 | 580 | 1094 | 24 | 1968 - 1987 | 1987 - 2006 | Massif Central |
| 12 | Creuse@Age | 559016 | 2144605 | 1121.8 | 46.0 | 266 | 553 | 948 | 3.57 | 1072 | 650 | 416 | 24 | 1960 - 1983 | 1983 - 2005 | Massif Central |
| 13 | Dordogne@Bort | 612697 | 2046020 | 1012.6 | 45.4 | 528 | 836 | 1774 | 4.15 | 1327 | 621 | 724 | 24 | 1979 - 1992 | 1992 - 2005 | Massif Central |
| 14 | Doubs@Brenet | 931797 | 2240815 | 941.0 | 47.4 | 768 | 1000 | 1438 | 5.09 | 1445 | 698 | 822 | 24 | 1962 - 1983 | 1983 - 2004 | Jura |
| 15 | Doubs@Neublans | 829420 | 2218279 | 7366.2 | 46.9 | 184 | 580 | 1438 | 4.55 | 1422 | 632 | 764 | 24 | 1967 - 1984 | 1984 - 2001 | Jura |
| 16 | Drac@Sautet | 882507 | 1985816 | 990.6 | 44.5 | 780 | 1742 | 3512 | 20.84 | 1421 | 579 | 1049 | 24 | 1969 - 1984 | 1984 - 1999 | Alps |
| 17 | Durance@Clapiere | 930627 | 1959457 | 2174.2 | 44.6 | 790 | 2113 | 3980 | 21.56 | 1159 | 444 | 725 | 24 | 1982 - 1994 | 1994 - 2005 | Alps |
| 18 | Eyrieux@Pontpierre | 785116 | 1983533 | 770.1 | 44.8 | 141 | 797 | 1710 | 10.98 | 1325 | 711 | 625 | 24 | 1998 - 2006 | 2006 - 2013 | Massif Central |
| 19 | Gage@GageIl | 739315 | 1979094 | 42.4 | 44.5 | 1011 | 1254 | 1592 | 6.36 | 1506 | 586 | 1107 | 6 | 1992 - 1999 | 1999 - 2005 | Massif Central |
| 20 | Gardon@Corbes | 730274 | 1898530 | 262.3 | 44.1 | 145 | 531 | 1181 | 12.02 | 1369 | 729 | 801 | 8 | 1986 - 1997 | 1997 - 2007 | Massif Central |
| 21 | Gardon@Generargues | 730935 | 1898983 | 245.2 | 44.1 | 166 | 562 | 1124 | 12.81 | 1372 | 770 | 752 | 8 | 1986 - 1997 | 1997 - 2008 | Massif Central |
| 22 | GaveEstaube@Gloriettes | 412443 | 1752747 | 20.2 | 42.4 | 1648 | 2192 | 2849 | 25.87 | 2001 | 673 | 1635 | 12 | 1987 - 1997 | 1997 - 2006 | Pyrenees |
| 23 | Goussant@PontRolland | 233822 | 2402559 | 422.5 | 48.5 | 31 | 109 | 338 | 1.54 | 824 | 685 | 224 | 24 | 1993 - 1999 | 1999 - 2004 | Britany region |
| 24 | Loire@Basset | 739771 | 2034325 | 3261.8 | 45.3 | 446 | 965 | 1701 | 5.49 | 993 | 731 | 419 | 24 | 1956 - 1983 | 1983 - 2010 | Massif Central |
| 25 | Loire@LaPalisse | 738771 | 1977633 | 130.9 | 44.5 | 1003 | 1245 | 1570 | 6.02 | 1778 | 585 | 1398 | 6 | 1992 - 1998 | 1998 - 2004 | Massif Central |
| 26 | Lot@Castelnau | 642336 | 1945506 | 1628.2 | 44.3 | 416 | 981 | 1700 | 7.24 | 1042 | 629 | 457 | 24 | 1982 - 1994 | 1994 - 2005 | Massif Central |
| 27 | Mimente@Florac | 701260 | 1924715 | 126.1 | 44.0 | 559 | 933 | 1409 | 11.09 | 1453 | 650 | 840 | 24 | 1985 - 1996 | 1996 - 2006 | Massif Central |
| 28 | Montane@Eyrein | 568891 | 2038403 | 43.2 | 45.3 | 553 | 624 | 803 | 3.09 | 1497 | 662 | 931 | 24 | 1958 - 1979 | 1979 - 1999 | Massif Central |
| 29 | Oriege@Campauleil | 560272 | 1745078 | 89.4 | 42.7 | 828 | 1829 | 2748 | 24.50 | 1928 | 479 | 1522 | 24 | 1960 - 1971 | 1971 - 1982 | Pyrenees |
| 30 | Ouveze@Bedarrides | 805366 | 1896400 | 1841.9 | 44.2 | 25 | 510 | 1895 | 6.70 | 868 | 798 | 364 | 24 | 1996 - 2000 | 2000 - 2004 | Alps |
| 31 | Rizzanese@Barrage | 1165253 | 1660880 | 116.2 | 41.6 | 554 | 1086 | 2028 | 11.92 | 1309 | 892 | 778 | 6 | 1993 - 2000 | 2000 - 2007 | Corsica island |
| 32 | Romanche@Chambon | 899366 | 2012160 | 256.6 | 45.2 | 1063 | 2365 | 3797 | 24.19 | 1468 | 427 | 1020 | 24 | 1985 - 1995 | 1995 - 2005 | Alps |
| 33 | Roya@Breil | 1015841 | 1895339 | 444.4 | 43.5 | 316 | 1475 | 2893 | 21.48 | 1307 | 650 | 776 | 24 | 1987 - 1996 | 1996 - 2005 | Alps |
| 34 | Salat@Roquefort | 488576 | 1795992 | 1580.2 | 42.9 | 269 | 982 | 2807 | 15.43 | 1468 | 713 | 786 | 24 | 1993 - 1998 | 1998 - 2003 | Pyrenees |
| 35 | Sioule@Fades | 635772 | 2108190 | 1290.6 | 45.9 | 508 | 778 | 1508 | 3.52 | 1101 | 727 | 435 | 24 | 1956 - 1982 | 1982 - 2008 | Massif Central |
| 36 | Souloise@Infernet | 882093 | 1980560 | 163.0 | 44.8 | 854 | 1685 | 2737 | 16.94 | 1636 | 677 | 1224 | 24 | 1991 - 1999 | 1999 - 2006 | Alps |
| 37 | Stura@Lanzo | 1003561 | 2042987 | 579.2 | 45.1 | 462 | 1785 | 3625 | 22.98 | 1497 | 478 | 1080 | 24 | 1954 - 1963 | 1963 - 1972 | Alps |
| 38 | Tarn@Cocures | 702310 | 1927978 | 189.4 | 44.0 | 583 | 1192 | 1697 | 9.11 | 1482 | 627 | 962 | 24 | 1985 - 1996 | 1996 - 2006 | Massif Central |
| 39 | Tarn@Millau | 658957 | 1899147 | 2085.7 | 44.0 | 349 | 896 | 1697 | 8.69 | 1213 | 642 | 630 | 24 | 1980 - 1995 | 1995 - 2010 | Massif Central |
| 40 | Tarn@Montbrun | 692563 | 1926599 | 588.6 | 44.0 | 487 | 1017 | 1700 | 10.55 | 1353 | 619 | 861 | 24 | 1985 - 1996 | 1996 - 2006 | Massif Central |
| 41 | Tarn@Pinet | 637927 | 1896585 | 2624.6 | 44.0 | 315 | 848 | 1697 | 8.29 | 1201 | 615 | 598 | 24 | 1980 - 1988 | 1988 - 1995 | Massif Central |
| 42 | Tarnon@Florac | 700826 | 1924522 | 133.1 | 44.0 | 557 | 980 | 1550 | 11.74 | 1392 | 620 | 813 | 24 | 1985 - 1996 | 1996 - 2006 | Massif Central |
| 43 | Taurion@RocheTalamie | 544532 | 2112334 | 654.1 | 46.0 | 379 | 562 | 883 | 3.40 | 1335 | 700 | 670 | 24 | 1957 - 1980 | 1980 - 2002 | Massif Central |
| 44 | Tech@Reynes | 627165 | 1718867 | 346.0 | 42.5 | 225 | 1132 | 2696 | 16.04 | 1097 | 685 | 518 | 24 | 1971 - 1978 | 1978 - 1985 | Pyrenees |
| 45 | Tet@Vinca | 569847 | 1734866 | 946.2 | 42.6 | 235 | 1394 | 2823 | 16.35 | 827 | 682 | 365 | 24 | 1982 - 1994 | 1994 - 2006 | Pyrenees |
| 46 | Tinee@PontLune | 988003 | 1893796 | 707.0 | 44.2 | 317 | 1750 | 2998 | 22.45 | 1260 | 563 | 643 | 24 | 1981 - 1992 | 1992 - 2002 | Alps |
| 47 | Truyere@Grandval | 658325 | 1991594 | 1788.8 | 44.5 | 723 | 1061 | 1500 | 4.07 | 997 | 581 | 454 | 24 | 1973 - 1990 | 1990 - 2006 | Massif Central |
| 48 | Ubaye@RocheRousse | 923493 | 1947189 | 946.6 | 44.4 | 804 | 2086 | 3317 | 20.78 | 1098 | 447 | 671 | 24 | 1982 - 1994 | 1994 - 2005 | Alps |
| 49 | Vence@Francheville | 771573 | 2528249 | 125.6 | 49.4 | 152 | 219 | 335 | 3.04 | 1022 | 653 | 577 | 24 | 1974 - 1985 | 1985 - 1995 | Ardennes region |
| 50 | Vienne@Bussy | 552421 | 2082789 | 379.2 | 46.9 | 407 | 691 | 951 | 4.64 | 1328 | 713 | 755 | 24 | 1983 - 1995 | 1995 - 2006 | Massif Central |

**S2 Model comparison over calibration periods**

This section details the performance of MORDOR V0, V1 and V2 over the calibration periods P1 and P2. For the three considered models, tables S2 and S3 show the values of the individual associated calibration metrics (KGE(Q), KGE(Qsea) and KGE(FDC)) for all the catchments over the calibration period P1 (table S2) and the calibration period P2 (table S3). The following figures show, for the 50 catchments, the observed hydrograph (year by year) and those modeled by MORDOR V0, V1 and SD (calibration mode).

**Table S2:** The values of the individual associated calibration metrics (KGE(Q), KGE(Qsea) and KGE(FDC)) for MORDOR V0, V1 and SD for all the catchments over the period P1

| ID | BV | MORDOR V0 | | | MORDOR V1 | | | MORDOR SD | | |
|---|---|---|---|---|---|---|---|---|---|---|
| | | KGE(Q) | KGE(Qsea) | KGE(FDC) | KGE(Q) | KGE(Qsea) | KGE(FDC) | KGE(Q) | KGE(Qsea) | KGE(FDC) |
| 1 | Agout@Fraisse | 0.948 | 0.995 | 0.898 | 0.963 | 0.991 | 0.914 | 0.955 | 0.974 | 0.900 |
| 2 | Agout@LaRaviege | 0.911 | 0.978 | 0.953 | 0.910 | 0.973 | 0.950 | 0.906 | 0.969 | 0.954 |
| 3 | Allier@Poutes | 0.960 | 0.889 | 0.925 | 0.969 | 0.885 | 0.939 | 0.960 | 0.881 | 0.941 |
| 4 | Ardeche@Sauze | 0.928 | 0.958 | 0.924 | 0.951 | 0.964 | 0.923 | 0.948 | 0.964 | 0.922 |
| 5 | Arn@Taillades | 0.885 | 0.938 | 0.987 | 0.909 | 0.944 | 0.989 | 0.923 | 0.938 | 0.989 |
| 6 | Arve@Arthaz | 0.984 | 0.817 | 0.918 | 0.991 | 0.812 | 0.926 | 0.981 | 0.792 | 0.929 |
| 7 | Behine@LaPoutroie | 0.961 | 0.962 | 0.903 | 0.957 | 0.965 | 0.882 | 0.954 | 0.939 | 0.908 |
| 8 | Breze@Meyrueis | 0.879 | 0.926 | 0.984 | 0.847 | 0.943 | 0.983 | 0.853 | 0.939 | 0.984 |
| 9 | Bromme@Brommat | 0.984 | 0.841 | 0.943 | 0.985 | 0.880 | 0.940 | 0.986 | 0.883 | 0.939 |
| 10 | Ceze@Bagnols | 0.947 | 0.943 | 0.887 | 0.954 | 0.917 | 0.892 | 0.959 | 0.940 | 0.880 |
| 11 | Chassezac@SteMarguerite | 0.903 | 0.939 | 0.959 | 0.924 | 0.944 | 0.956 | 0.911 | 0.950 | 0.957 |
| 12 | Creuse@Age | 0.980 | 0.888 | 0.946 | 0.983 | 0.901 | 0.948 | 0.984 | 0.896 | 0.944 |
| 13 | Dordogne@Bort | 0.913 | 0.924 | 0.889 | 0.926 | 0.929 | 0.879 | 0.934 | 0.927 | 0.868 |
| 14 | Doubs@Brenet | 0.874 | 0.909 | 0.976 | 0.901 | 0.910 | 0.985 | 0.874 | 0.920 | 0.981 |
| 15 | Doubs@Neublans | 0.974 | 0.898 | 0.962 | 0.981 | 0.874 | 0.967 | 0.981 | 0.895 | 0.966 |
| 16 | Drac@Sautet | 0.940 | 0.974 | 0.916 | 0.927 | 0.956 | 0.901 | 0.933 | 0.985 | 0.899 |
| 17 | Durance@Clapiere | 0.900 | 0.926 | 0.975 | 0.895 | 0.933 | 0.983 | 0.903 | 0.939 | 0.981 |
| 18 | Eyrieux@Pontpierre | 0.982 | 0.848 | 0.951 | 0.979 | 0.862 | 0.963 | 0.978 | 0.843 | 0.961 |
| 19 | Gage@Gagell | 0.931 | 0.971 | 0.921 | 0.915 | 0.974 | 0.937 | 0.894 | 0.970 | 0.924 |
| 20 | Gardon@Corbes | 0.882 | 0.906 | 0.984 | 0.832 | 0.919 | 0.982 | 0.787 | 0.935 | 0.982 |
| 21 | Gardon@Generargues | 0.968 | 0.909 | 0.928 | 0.967 | 0.921 | 0.958 | 0.960 | 0.906 | 0.957 |
| 22 | GaveEstaube@Gloriettes | 0.933 | 0.979 | 0.871 | 0.942 | 0.984 | 0.886 | 0.935 | 0.980 | 0.871 |
| 23 | Goussant@PontRolland | 0.849 | 0.947 | 0.967 | 0.841 | 0.962 | 0.961 | 0.823 | 0.958 | 0.965 |
| 24 | Loire@Basset | 0.966 | 0.815 | 0.931 | 0.969 | 0.832 | 0.935 | 0.969 | 0.815 | 0.935 |
| 25 | Loire@LaPalisse | 0.940 | 0.943 | 0.884 | 0.937 | 0.944 | 0.896 | 0.912 | 0.958 | 0.891 |
| 26 | Lot@Castelnau | 0.838 | 0.946 | 0.958 | 0.853 | 0.949 | 0.959 | 0.826 | 0.950 | 0.961 |
| 27 | Mimente@Florac | 0.954 | 0.925 | 0.942 | 0.967 | 0.951 | 0.947 | 0.948 | 0.959 | 0.953 |
| 28 | Montane@Eyrein | 0.918 | 0.975 | 0.895 | 0.926 | 0.982 | 0.917 | 0.928 | 0.982 | 0.929 |
| 29 | Oriege@Campauleil | 0.820 | 0.944 | 0.986 | 0.819 | 0.953 | 0.989 | 0.826 | 0.951 | 0.987 |
| 30 | Ouveze@Bedarrides | 0.971 | 0.863 | 0.958 | 0.977 | 0.926 | 0.946 | 0.978 | 0.929 | 0.942 |
| 31 | Rizzanese@Barrage | 0.948 | 0.971 | 0.897 | 0.951 | 0.973 | 0.854 | 0.943 | 0.973 | 0.815 |
| 32 | Romanche@Chambon | 0.932 | 0.948 | 0.970 | 0.941 | 0.956 | 0.975 | 0.912 | 0.945 | 0.971 |
| 33 | Roya@Breil | 0.975 | 0.939 | 0.879 | 0.978 | 0.931 | 0.887 | 0.977 | 0.913 | 0.884 |
| 34 | Salat@Roquefort | 0.952 | 0.990 | 0.872 | 0.937 | 0.991 | 0.892 | 0.953 | 0.991 | 0.899 |
| 35 | Sioule@Fades | 0.924 | 0.906 | 0.957 | 0.928 | 0.906 | 0.969 | 0.930 | 0.903 | 0.969 |
| 36 | Souloise@Infernet | 0.992 | 0.883 | 0.918 | 0.992 | 0.869 | 0.910 | 0.991 | 0.858 | 0.949 |
| 37 | Stura@Lanzo | 0.916 | 0.988 | 0.841 | 0.951 | 0.988 | 0.796 | 0.945 | 0.986 | 0.862 |
| 38 | Tarn@Cocures | 0.904 | 0.927 | 0.932 | 0.930 | 0.934 | 0.946 | 0.935 | 0.931 | 0.954 |
| 39 | Tarn@Millau | 0.981 | 0.793 | 0.927 | 0.984 | 0.799 | 0.938 | 0.985 | 0.806 | 0.934 |
| 40 | Tarn@Montbrun | 0.921 | 0.973 | 0.926 | 0.936 | 0.980 | 0.942 | 0.918 | 0.977 | 0.924 |
| 41 | Tarn@Pinet | 0.917 | 0.922 | 0.982 | 0.922 | 0.902 | 0.980 | 0.923 | 0.900 | 0.981 |
| 42 | Tarnon@Florac | 0.962 | 0.775 | 0.946 | 0.972 | 0.731 | 0.946 | 0.966 | 0.806 | 0.957 |
| 43 | Taurion@RocheTalamie | 0.934 | 0.956 | 0.906 | 0.956 | 0.954 | 0.885 | 0.951 | 0.948 | 0.906 |
| 44 | Tech@Reynes | 0.929 | 0.937 | 0.993 | 0.950 | 0.939 | 0.991 | 0.955 | 0.949 | 0.976 |
| 45 | Tet@Vinca | 0.985 | 0.896 | 0.932 | 0.990 | 0.876 | 0.935 | 0.987 | 0.855 | 0.934 |
| 46 | Tinee@PontLune | 0.924 | 0.989 | 0.879 | 0.949 | 0.983 | 0.873 | 0.958 | 0.994 | 0.840 |
| 47 | Truyere@Grandval | 0.899 | 0.864 | 0.985 | 0.901 | 0.899 | 0.987 | 0.923 | 0.895 | 0.987 |
| 48 | Ubaye@RocheRousse | 0.991 | 0.798 | 0.957 | 0.990 | 0.802 | 0.953 | 0.966 | 0.832 | 0.962 |
| 49 | Vence@Francheville | 0.945 | 0.919 | 0.903 | 0.934 | 0.960 | 0.931 | 0.966 | 0.951 | 0.924 |
| 50 | Vienne@Bussy | 0.871 | 0.900 | 0.980 | 0.846 | 0.904 | 0.985 | 0.887 | 0.906 | 0.982 |

**Table S3:** The values of the individual associated calibration metrics (KGE(Q), KGE(Qsea) and KGE(FDC)) for MORDOR V0, V1 and SD for all the catchments over the period P3

| ID | BV | MORDOR V0 | | | MORDOR V1 | | | MORDOR SD | | |
|---|---|---|---|---|---|---|---|---|---|---|
| | | KGE(Q) | KGE(Qsea) | KGE(FDC) | KGE(Q) | KGE(Qsea) | KGE(FDC) | KGE(Q) | KGE(Qsea) | KGE(FDC) |
| 1 | Agout@Fraisse | 0.977 | 0.846 | 0.922 | 0.981 | 0.833 | 0.921 | 0.976 | 0.842 | 0.932 |
| 2 | Agout@LaRaviege | 0.938 | 0.981 | 0.880 | 0.947 | 0.982 | 0.871 | 0.941 | 0.980 | 0.910 |
| 3 | Allier@Poutes | 0.878 | 0.940 | 0.986 | 0.901 | 0.951 | 0.988 | 0.869 | 0.950 | 0.988 |
| 4 | Ardeche@Sauze | 0.981 | 0.866 | 0.937 | 0.986 | 0.861 | 0.941 | 0.987 | 0.860 | 0.945 |
| 5 | Arn@Taillades | 0.962 | 0.958 | 0.916 | 0.973 | 0.963 | 0.907 | 0.969 | 0.958 | 0.900 |
| 6 | Arve@Arthaz | 0.908 | 0.932 | 0.975 | 0.931 | 0.939 | 0.972 | 0.857 | 0.919 | 0.978 |
| 7 | Behine@LaPoutroie | 0.963 | 0.826 | 0.950 | 0.960 | 0.828 | 0.948 | 0.953 | 0.785 | 0.961 |
| 8 | Breze@Meyrueis | 0.966 | 0.951 | 0.905 | 0.975 | 0.963 | 0.901 | 0.976 | 0.964 | 0.884 |
| 9 | Bromme@Brommat | 0.893 | 0.910 | 0.978 | 0.891 | 0.900 | 0.979 | 0.903 | 0.918 | 0.976 |
| 10 | Ceze@Bagnols | 0.971 | 0.801 | 0.915 | 0.977 | 0.770 | 0.917 | 0.977 | 0.820 | 0.927 |
| 11 | Chassezac@SteMarguerite | 0.937 | 0.963 | 0.788 | 0.955 | 0.970 | 0.783 | 0.958 | 0.971 | 0.795 |
| 12 | Creuse@Age | 0.908 | 0.895 | 0.958 | 0.923 | 0.898 | 0.955 | 0.910 | 0.901 | 0.953 |
| 13 | Dordogne@Bort | 0.956 | 0.788 | 0.947 | 0.975 | 0.801 | 0.967 | 0.974 | 0.804 | 0.963 |
| 14 | Doubs@Brenet | 0.907 | 0.982 | 0.898 | 0.934 | 0.967 | 0.922 | 0.928 | 0.980 | 0.926 |
| 15 | Doubs@Neublans | 0.910 | 0.897 | 0.978 | 0.917 | 0.893 | 0.977 | 0.905 | 0.923 | 0.977 |
| 16 | Drac@Sautet | 0.983 | 0.844 | 0.941 | 0.962 | 0.840 | 0.961 | 0.971 | 0.867 | 0.958 |
| 17 | Durance@Clapiere | 0.939 | 0.970 | 0.822 | 0.941 | 0.972 | 0.854 | 0.938 | 0.971 | 0.859 |
| 18 | Eyrieux@Pontpierre | 0.886 | 0.948 | 0.967 | 0.887 | 0.948 | 0.977 | 0.874 | 0.929 | 0.974 |
| 19 | Gage@GageII | 0.956 | 0.861 | 0.965 | 0.943 | 0.865 | 0.960 | 0.922 | 0.859 | 0.965 |
| 20 | Gardon@Corbes | 0.949 | 0.975 | 0.920 | 0.947 | 0.974 | 0.923 | 0.942 | 0.976 | 0.925 |
| 21 | Gardon@Generargues | 0.893 | 0.935 | 0.968 | 0.898 | 0.930 | 0.979 | 0.900 | 0.940 | 0.974 |
| 22 | GaveEstaube@Gloriettes | 0.952 | 0.932 | 0.948 | 0.960 | 0.916 | 0.954 | 0.955 | 0.899 | 0.954 |
| 23 | Goussant@PontRolland | 0.947 | 0.971 | 0.896 | 0.952 | 0.973 | 0.891 | 0.951 | 0.971 | 0.896 |
| 24 | Loire@Basset | 0.894 | 0.929 | 0.948 | 0.905 | 0.925 | 0.955 | 0.882 | 0.942 | 0.957 |
| 25 | Loire@LaPalisse | 0.954 | 0.756 | 0.943 | 0.945 | 0.745 | 0.944 | 0.915 | 0.799 | 0.944 |
| 26 | Lot@Castelnau | 0.923 | 0.981 | 0.873 | 0.933 | 0.986 | 0.863 | 0.927 | 0.988 | 0.891 |
| 27 | Mimente@Florac | 0.889 | 0.932 | 0.987 | 0.902 | 0.958 | 0.991 | 0.850 | 0.948 | 0.992 |
| 28 | Montane@Eyrein | 0.940 | 0.855 | 0.951 | 0.947 | 0.856 | 0.956 | 0.947 | 0.849 | 0.956 |
| 29 | Oriege@Campauleil | 0.964 | 0.954 | 0.902 | 0.972 | 0.966 | 0.898 | 0.975 | 0.963 | 0.890 |
| 30 | Ouveze@Bedarrides | 0.862 | 0.959 | 0.957 | 0.880 | 0.959 | 0.947 | 0.868 | 0.961 | 0.932 |
| 31 | Rizzanese@Barrage | 0.971 | 0.867 | 0.956 | 0.975 | 0.881 | 0.962 | 0.963 | 0.891 | 0.960 |
| 32 | Romanche@Chambon | 0.949 | 0.991 | 0.896 | 0.957 | 0.992 | 0.892 | 0.956 | 0.987 | 0.894 |
| 33 | Roya@Breil | 0.906 | 0.942 | 0.940 | 0.914 | 0.952 | 0.957 | 0.893 | 0.954 | 0.951 |
| 34 | Salat@Roquefort | 0.993 | 0.925 | 0.900 | 0.991 | 0.919 | 0.923 | 0.993 | 0.916 | 0.926 |
| 35 | Sioule@Fades | 0.954 | 0.987 | 0.895 | 0.962 | 0.984 | 0.891 | 0.962 | 0.987 | 0.901 |
| 36 | Souloise@Infernet | 0.921 | 0.901 | 0.984 | 0.944 | 0.905 | 0.981 | 0.922 | 0.892 | 0.991 |
| 37 | Stura@Lanzo | 0.969 | 0.903 | 0.881 | 0.986 | 0.908 | 0.899 | 0.986 | 0.884 | 0.907 |
| 38 | Tarn@Cocures | 0.948 | 0.944 | 0.851 | 0.951 | 0.946 | 0.825 | 0.956 | 0.951 | 0.823 |
| 39 | Tarn@Millau | 0.926 | 0.945 | 0.983 | 0.926 | 0.963 | 0.985 | 0.934 | 0.959 | 0.984 |
| 40 | Tarn@Montbrun | 0.971 | 0.810 | 0.928 | 0.975 | 0.855 | 0.930 | 0.972 | 0.844 | 0.934 |
| 41 | Tarn@Pinet | 0.930 | 0.936 | 0.916 | 0.929 | 0.925 | 0.920 | 0.919 | 0.931 | 0.914 |
| 42 | Tarnon@Florac | 0.908 | 0.899 | 0.990 | 0.917 | 0.904 | 0.989 | 0.902 | 0.891 | 0.982 |
| 43 | Taurion@RocheTalamie | 0.982 | 0.792 | 0.931 | 0.990 | 0.811 | 0.945 | 0.988 | 0.848 | 0.950 |
| 44 | Tech@Reynes | 0.956 | 0.984 | 0.867 | 0.967 | 0.990 | 0.859 | 0.967 | 0.984 | 0.878 |
| 45 | Tet@Vinca | 0.952 | 0.941 | 0.974 | 0.952 | 0.939 | 0.977 | 0.943 | 0.957 | 0.965 |
| 46 | Tinee@PontLune | 0.988 | 0.868 | 0.954 | 0.988 | 0.839 | 0.962 | 0.992 | 0.881 | 0.962 |
| 47 | Truyere@Grandval | 0.949 | 0.888 | 0.933 | 0.948 | 0.908 | 0.941 | 0.950 | 0.920 | 0.939 |
| 48 | Ubaye@RocheRousse | 0.923 | 0.886 | 0.980 | 0.914 | 0.930 | 0.986 | 0.904 | 0.928 | 0.988 |
| 49 | Vence@Francheville | 0.991 | 0.787 | 0.926 | 0.985 | 0.803 | 0.934 | 0.994 | 0.799 | 0.931 |
| 50 | Vienne@Bussy | 0.934 | 0.944 | 0.906 | 0.943 | 0.949 | 0.920 | 0.950 | 0.946 | 0.922 |

**Agout@Fraisse**

[Figure]

**Agout@Fraisse**

[Figure]

**Agout@Fraisse**

[Figure]

**Agout@LaRaviege**

[Figure]

**Agout@LaRaviege**

[Figure]

**Agout@LaRaviege**

[Figure]

**Year 1986**

**Year 1987**

[Figure]

Allier@Poutes

[Figure]

**Allier@Poutes**

**Allier@Poutes**

[Figure]

**Ardeche@Sauze**

[Figure]

[Figure]

**Ardeche@Sauze**

**Year 1968**

**Year 1969**

**Year 1970**

**Year 1971**

**Year 1972**

**Year 1973**

**Ardeche@Sauze**

[Figure]

**Ardeche@Sauze**

**Year 1980**

**Year 1981**

**Year 1982**

**Year 1983**

**Year 1984**

**Year 1985**

**Ardeche@Sauze**

[Figure]

**Ardeche@Sauze**

[Figure]

**Ardeche@Sauze**

[Figure]

**Ardeche@Sauze**

[Figure]

[Figure]

**Arn@Taillades**

**Arn@Taillades**

[Figure]

[Figure]

**Arn@Taillades**

**Arn@Taillades**

[Figure]

**Arve@Arthaz**

[Figure]

**Arve@Arthaz**

[Figure]

**Arve@Arthaz**

**Year 1999**

**Year 2000**

[Figure]

[Figure]

**Behine@LaPoutroie**

**Behine@LaPoutroie**

Year 1991

Year 1992

Year 1993

**Breze@Meyrueis**

[Figure]

**Breze@Meyrueis**

**Year 1992**

**Year 1993**

**Year 1994**

**Year 1995**

**Year 1996**

**Year 1997**

**Breze@Meyrueis**

[Figure]

**Breze@Meyrueis**

**Bromme@Brommat**

[Figure]

**Bromme@Brommat**

[Figure]

[Figure]

Ceze@Bagnols

**Ceze@Bagnols**

[Figure]

Chassezac@SteMarguerite

[Figure]

Chassezac@SteMarguerite

**Chassezac@SteMarguerite**

[Figure]

**Chassezac@SteMarguerite**

[Figure]

[Figure]

Chassezac@SteMarguerite

**Chassezac@SteMarguerite**

[Figure]

**Chassezac@SteMarguerite**

[Figure]

**Creuse@Age**

**Year 1961**

**Year 1962**

**Year 1963**

**Year 1964**

**Year 1965**

**Year 1966**

**Creuse@Age**

[Figure]

**Creuse@Age**

[Figure]

[Figure]

Creuse@Age

[Figure]

Creuse@Age

**Creuse@Age**

[Figure]

**Creuse@Age**

[Figure]

**Creuse@Age**

[Figure]

**Dordogne@Bort**

[Figure]

**Dordogne@Bort**

[Figure]

[Figure]

**Dordogne@Bort**

[Figure]

**Dordogne@Bort**

**Dordogne@Bort**

[Figure]

**Year 2004**

**Year 2005**

**Doubs@Brenet**

[Figure]

**Doubs@Brenet**

[Figure]

**Doubs@Brenet**

[Figure]

**Doubs@Brenet**

[Figure]

[Figure]

**Doubs@Brenet**

[Figure]

Doubs@Brenet

**Doubs@Brenet**

[Figure]

**Doubs@Neublans**

[Figure]

**Doubs@Neublans**

[Figure]

**Doubs@Neublans**

[Figure]

**Doubs@Neublans**

[Figure]

**Doubs@Neublans**

[Figure]

**Doubs@Neublans**

[Figure]

**Drac@Sautet**

[Figure]

[Figure]

[Figure]

[Figure]

**Drac@Sautet**

**Drac@Sautet**

[Figure]

**Durance@Clapiere**

[Figure]

[Figure]

Durance@Clapiere

**Durance@Clapiere**

[Figure]

**Durance@Clapiere**

[Figure]

**Eyrieux@Pontpierre**

[Figure]

**Eyrieux@Pontpierre**

**Eyrieux@Pontpierre**

**Gage@GageII**

[Figure]

**Gage@GageII**

[Figure]

[Figure]

[Figure]

Gardon@Corbes

**Gardon@Corbes**

[Figure]

**Gardon@Corbes**

[Figure]

**Gardon@Corbes**

[Figure]

**Gardon@Generargues**

[Figure]

**Gardon@Generargues**

[Figure]

**Gardon@Generargues**

[Figure]

**Gardon@Generargues**

[Figure]

**GaveEstaube@Gloriettes**

[Figure]

**GaveEstaube@Gloriettes**

[Figure]

**GaveEstaube@Gloriettes**

[Figure]

[Figure]

**Goussant@PontRolland**

**Goussant@PontRolland**

**Loire@Basset**

[Figure]

**Loire@Basset**

[Figure]

**Loire@Basset**

[Figure]

**Loire@Basset**

[Figure]

**Loire@Basset**

[Figure]

[Figure]

[Figure]

Loire@Basset

**Loire@Basset**

[Figure]

**Loire@Basset**

[Figure]

**Loire@LaPalisse**

[Figure]

[Figure]

Loire@LaPalisse

[Figure]

Lot@Castelnau

[Figure]

Lot@Castelnau

**Lot@Castelnau**

**Year 1995**

**Year 1996**

**Year 1997**

**Year 1998**

**Year 1999**

**Year 2000**

**Lot@Castelnau**

**Year 2001**

**Year 2002**

**Year 2003**

**Year 2004**

**Year 2005**

[Figure]

**Mimente@Florac**

[Figure]

**Mimente@Florac**

[Figure]

**Mimente@Florac**

[Figure]

**Mimente@Florac**

[Figure]

**Montane@Eyrein**

[Figure]

**Montane@Eyrein**

[Figure]

[Figure]

**Montane@Eyrein**

[Figure]

**Montane@Eyrein**

**Montane@Eyrein**

[Figure]

[Figure]

**Montane@Eyrein**

**Montane@Eyrein**

[Figure]

**Oriege@Campauleil**

[Figure]

**Oriege@Campauleil**

[Figure]

[Figure]

Oriege@Campauleil

**Oriege@Campauleil**

[Figure]

[Figure]

Ouveze@Bedarrides

**Ouveze@Bedarrides**

**Year 2003**

**Year 2004**

**Rizzanese@Barrage**

**Rizzanese@Barrage**

**Rizzanese@Barrage**

**Year 2006**

**Year 2007**

**Romanche@Chambon**

[Figure]

**Romanche@Chambon**

[Figure]

[Figure]

**Romanche@Chambon**

**Romanche@Chambon**

**Year 2004**

**Year 2005**

[Figure]

**Roya@Breil**

**Roya@Breil**

[Figure]

[Figure]

[Figure]

**Salat@Roquefort**

[Figure]

**Sioule@Fades**

[Figure]

**Sioule@Fades**

[Figure]

**Year 1963**

**Year 1964**

**Year 1965**

**Year 1966**

**Year 1967**

**Year 1968**

**Sioule@Fades**

[Figure]

**Year 1969**

**Year 1970**

**Year 1971**

**Year 1972**

**Year 1973**

**Year 1974**

**Sioule@Fades**

[Figure]

**Sioule@Fades**

[Figure]

**Sioule@Fades**

[Figure]

**Sioule@Fades**

[Figure]

[Figure]

**Sioule@Fades**

**Sioule@Fades**

[Figure]

**Souloise@Infernet**

[Figure]

**Souloise@Infernet**

[Figure]

**Souloise@Infernet**

[Figure]

[Figure]

Stura@Lanzo

**Stura@Lanzo**

**Tarn@Cocures**

[Figure]

**Tarn@Cocures**

[Figure]

**Tarn@Cocures**

[Figure]

**Tarn@Cocures**

[Figure]

**Tarn@Millau**

[Figure]

**Tarn@Millau**

[Figure]

**Tarn@Millau**

[Figure]

**Year 1993**

**Year 1994**

**Year 1995**

**Year 1996**

**Year 1997**

**Year 1998**

[Figure]

Tarn@Millau

[Figure]

**Tarn@Millau**

**Tarn@Montbrun**

[Figure]

**Tarn@Montbrun**

[Figure]

**Tarn@Montbrun**

[Figure]

**Tarn@Montbrun**

[Figure]

[Figure]

**Tarn@Pinet**

[Figure]

**Tarn@Pinet**

[Figure]

**Tarnon@Florac**

[Figure]

[Figure]

Tarnon@Florac

[Figure]

Tarnon@Florac

**Tarnon@Florac**

[Figure]

**Taurion@RocheTalamie**

[Figure]

[Figure]

**Taurion@RocheTalamie**

[Figure]

**Taurion@RocheTalamie**

**Taurion@RocheTalamie**

[Figure]

**Taurion@RocheTalamie**

[Figure]

**Taurion@RocheTalamie**

[Figure]

**Taurion@RocheTalamie**

[Figure]

**Taurion@RocheTalamie**

**Tech@Reynes**

[Figure]

**Tech@Reynes**

[Figure]

**Tech@Reynes**

[Figure]

[Figure]

**Tet@Vinca**

[Figure]

[Figure]

Tet@Vinca

[Figure]

Tet@Vinca

**Tinee@PontLune**

[Figure]

[Figure]

**Tinee@PontLune**

Tinee@PontLune

**Tinee@PontLune**

[Figure]

**Truyere@Grandval**

[Figure]

[Figure]

**Truyere@Grandval**

**Truyere@Grandval**

[Figure]

[Figure]

Truyere@Grandval

**Truyere@Grandval**

[Figure]

Truyere@Grandval

**Ubaye@RocheRousse**

[Figure]

[Figure]

Ubaye@RocheRousse

**Ubaye@RocheRousse**

[Figure]

**Ubaye@RocheRousse**

[Figure]

**Vence@Francheville**

[Figure]

**Vence@Francheville**

[Figure]

**Vence@Francheville**

[Figure]

**Vence@Francheville**

[Figure]

**Vienne@Bussy**

[Figure]

**Vienne@Bussy**

[Figure]

**Vienne@Bussy**

[Figure]

**Vienne@Bussy**

[Figure]

**S3 Model comparison over evaluation periods**

This section details the performance of MORDOR V0, V1 and V2 over the evaluation periods P1 and P2. For the three considered models, tables S4 and S5 show the values of the individual associated evaluation metrics (NSE(Q), NSE(Qsea), NSE(dQ), NSE(FDC) and NSE(Qlow)) for all the catchments over the evaluation period P1 (table S4) and the evaluation period P2 (table S5). The following figures show, for the 50 catchments, the observed hydrograph (year by year) and those modeled by MORDOR V0, V1 and SD (evaluation model).

**Table S4:** The values of the individual associated evaluation metrics (NSE(Q), NSE(Qsea), NSE(dQ), NSE(FDC) and NSE(Qlow)) for MORDOR V0, V1 and SD for all the catchments over the period P1

| ID | BV | MORDOR V0 | | | | | MORDOR V1 | | | | | MORDOR SD | | | | |
|----|----|-----------|--|--|--|--|-----------|--|--|--|--|-----------|--|--|--|--|
| | | NSE(Q) | NSE(Qsea) | NSE(dQ) | NSE(FDC) | NSE(Qlow) | NSE(Q) | NSE(Qsea) | NSE(dQ) | NSE(FDC) | NSE(Qlow) | NSE(Q) | NSE(Qsea) | NSE(dQ) | NSE(FDC) | NSE(Qlow) |
| 1 | Agout@Fraisse | 0.838 | 0.849 | 0.795 | 0.747 | 0.889 | 0.925 | 0.906 | 0.889 | 0.805 | 0.917 | 0.911 | 0.896 | 0.902 | 0.813 | 0.917 |
| 2 | Agout@LaRaviege | 0.342 | 0.483 | 0.452 | 0.237 | 0.701 | 0.699 | 0.725 | 0.633 | 0.406 | 0.786 | 0.706 | 0.727 | 0.693 | 0.499 | 0.789 |
| 3 | Allier@Poutes | 0.759 | 0.842 | 0.818 | 0.273 | 0.752 | 0.926 | 0.915 | 0.825 | 0.582 | 0.851 | 0.888 | 0.913 | 0.868 | 0.674 | 0.880 |
| 4 | Ardeche@Sauze | 0.937 | 0.935 | 0.858 | 0.875 | 0.883 | 0.938 | 0.959 | 0.860 | 0.909 | 0.927 | 0.923 | 0.954 | 0.855 | 0.896 | 0.933 |
| 5 | Arn@Taillades | 0.936 | 0.957 | 0.891 | 0.848 | 0.874 | 0.939 | 0.974 | 0.930 | 0.905 | 0.957 | 0.920 | 0.972 | 0.935 | 0.930 | 0.958 |
| 6 | Arve@Arthaz | 0.907 | 0.907 | 0.796 | 0.873 | 0.882 | 0.907 | 0.882 | 0.826 | 0.879 | 0.880 | 0.905 | 0.906 | 0.839 | 0.898 | 0.876 |
| 7 | Behine@LaPoutroie | 0.737 | 0.639 | 0.516 | -0.341 | 0.662 | 0.564 | 0.490 | 0.656 | 0.262 | 0.641 | 0.567 | 0.656 | 0.546 | 0.368 | 0.631 |
| 8 | Breze@Meyrueis | 0.889 | 0.802 | 0.442 | 0.282 | 0.779 | 0.860 | 0.840 | 0.602 | 0.602 | 0.857 | 0.863 | 0.864 | 0.553 | 0.777 | 0.874 |
| 9 | Bromme@Brommat | 0.948 | 0.982 | 0.923 | 0.967 | 0.831 | 0.984 | 0.986 | 0.944 | 0.965 | 0.919 | 0.965 | 0.985 | 0.971 | 0.989 | 0.922 |
| 10 | Ceze@Bagnols | 0.913 | 0.972 | 0.914 | 0.883 | 0.915 | 0.970 | 0.990 | 0.907 | 0.852 | 0.954 | 0.944 | 0.982 | 0.938 | 0.930 | 0.957 |
| 11 | Chassezac@SteMarguerite | 0.898 | 0.866 | 0.821 | 0.656 | 0.879 | 0.916 | 0.903 | 0.868 | 0.798 | 0.895 | 0.908 | 0.890 | 0.879 | 0.790 | 0.907 |
| 12 | Creuse@Age | 0.630 | 0.577 | 0.235 | 0.362 | 0.192 | 0.801 | 0.572 | 0.417 | 0.550 | 0.586 | 0.791 | 0.563 | 0.608 | 0.540 | 0.673 |
| 13 | Dordogne@Bort | 0.867 | 0.821 | 0.933 | 0.657 | 0.760 | 0.886 | 0.845 | 0.806 | 0.637 | 0.804 | 0.865 | 0.826 | 0.798 | 0.661 | 0.829 |
| 14 | Doubs@Brenet | 0.851 | 0.945 | 0.944 | 0.549 | 0.932 | 0.972 | 0.970 | 0.951 | 0.924 | 0.982 | 0.975 | 0.971 | 0.942 | 0.912 | 0.978 |
| 15 | Doubs@Neublans | 0.902 | 0.968 | 0.899 | 0.674 | 0.920 | 0.952 | 0.980 | 0.934 | 0.857 | 0.975 | 0.952 | 0.980 | 0.929 | 0.859 | 0.975 |
| 16 | Drac@Sautet | 0.843 | 0.828 | 0.792 | 0.815 | 0.783 | 0.907 | 0.873 | 0.837 | 0.808 | 0.896 | 0.918 | 0.839 | 0.871 | 0.811 | 0.888 |
| 17 | Durance@Clapiere | 0.631 | 0.597 | 0.459 | 0.616 | 0.268 | 0.801 | 0.640 | 0.540 | 0.628 | 0.867 | 0.842 | 0.491 | 0.699 | 0.578 | 0.844 |
| 18 | Eyrieux@Pontpierre | 0.748 | 0.756 | 0.874 | 0.701 | 0.706 | 0.802 | 0.812 | 0.835 | 0.489 | 0.827 | 0.829 | 0.756 | 0.868 | 0.433 | 0.678 |
| 19 | Gage@Gagell | 0.934 | 0.888 | 0.912 | 0.818 | 0.764 | 0.966 | 0.945 | 0.969 | 0.903 | 0.950 | 0.971 | 0.933 | 0.962 | 0.913 | 0.942 |
| 20 | Gardon@Corbes | 0.953 | 0.969 | 0.952 | 0.873 | 0.886 | 0.970 | 0.972 | 0.979 | 0.940 | 0.934 | 0.968 | 0.969 | 0.977 | 0.950 | 0.943 |
| 21 | Gardon@Generargues | 0.803 | 0.871 | 0.840 | 0.863 | 0.798 | 0.854 | 0.914 | 0.926 | 0.880 | 0.776 | 0.870 | 0.904 | 0.921 | 0.884 | 0.767 |
| 22 | GaveEstaube@Gloriettes | 0.701 | 0.666 | 0.477 | 0.580 | 0.408 | 0.480 | 0.685 | 0.729 | 0.607 | 0.339 | 0.483 | 0.644 | 0.703 | 0.670 | 0.293 |
| 23 | Goussant@PontRolland | 0.849 | 0.773 | 0.634 | 0.850 | 0.487 | 0.784 | 0.825 | 0.815 | 0.886 | 0.509 | 0.904 | 0.813 | 0.829 | 0.901 | 0.674 |
| 24 | Loire@Basset | 0.909 | 0.896 | 0.940 | 0.973 | 0.825 | 0.964 | 0.980 | 0.905 | 0.979 | 0.940 | 0.963 | 0.974 | 0.930 | 0.980 | 0.940 |
| 25 | Loire@LaPalisse | 0.871 | 0.960 | 0.910 | 0.964 | 0.877 | 0.959 | 0.988 | 0.869 | 0.970 | 0.953 | 0.952 | 0.984 | 0.902 | 0.969 | 0.952 |
| 26 | Lot@Castelnau | 0.865 | 0.811 | 0.879 | 0.866 | 0.384 | 0.859 | 0.900 | 0.901 | 0.851 | 0.830 | 0.866 | 0.919 | 0.901 | 0.859 | 0.904 |
| 27 | Mimente@Florac | 0.618 | 0.391 | 0.504 | 0.688 | -0.310 | 0.684 | 0.634 | 0.640 | 0.686 | 0.457 | 0.736 | 0.666 | 0.616 | 0.627 | 0.774 |
| 28 | Montane@Eyrein | 0.757 | 0.744 | 0.914 | 0.644 | 0.898 | 0.863 | 0.702 | 0.908 | 0.630 | 0.478 | 0.877 | 0.830 | 0.903 | 0.761 | 0.685 |
| 29 | Oriege@Campauleil | 0.958 | 0.977 | 0.898 | 0.928 | 0.899 | 0.957 | 0.981 | 0.965 | 0.967 | 0.895 | 0.956 | 0.967 | 0.965 | 0.968 | 0.913 |
| 30 | Ouveze@Bedarrides | 0.948 | 0.955 | 0.890 | 0.941 | 0.560 | 0.951 | 0.964 | 0.955 | 0.948 | 0.859 | 0.953 | 0.960 | 0.947 | 0.951 | 0.884 |
| 31 | Rizzanese@Barrage | 0.880 | 0.851 | 0.880 | 0.811 | 0.857 | 0.838 | 0.873 | 0.915 | 0.882 | 0.878 | 0.799 | 0.939 | 0.906 | 0.882 | 0.876 |
| 32 | Romanche@Chambon | 0.574 | 0.209 | 0.620 | 0.605 | 0.460 | 0.569 | 0.366 | 0.709 | 0.535 | 0.601 | 0.356 | 0.528 | 0.693 | 0.541 | 0.591 |
| 33 | Roya@Breil | 0.457 | 0.716 | 0.674 | 0.342 | 0.898 | 0.804 | 0.577 | 0.890 | 0.584 | 0.906 | 0.803 | 0.589 | 0.874 | 0.657 | 0.908 |
| 34 | Salat@Roquefort | 0.885 | 0.985 | 0.898 | 0.890 | 0.934 | 0.937 | 0.982 | 0.945 | 0.912 | 0.961 | 0.925 | 0.984 | 0.947 | 0.920 | 0.964 |
| 35 | Sioule@Fades | 0.900 | 0.860 | 0.969 | 0.770 | 0.961 | 0.936 | 0.898 | 0.950 | 0.840 | 0.974 | 0.937 | 0.923 | 0.954 | 0.859 | 0.967 |
| 36 | Souloise@Infernet | 0.889 | 0.865 | 0.892 | 0.854 | 0.881 | 0.884 | 0.949 | 0.913 | 0.896 | 0.898 | 0.871 | 0.945 | 0.890 | 0.889 | 0.916 |
| 37 | Stura@Lanzo | 0.756 | 0.504 | 0.564 | 0.449 | 0.454 | 0.641 | 0.880 | 0.640 | 0.687 | 0.509 | 0.640 | 0.876 | 0.553 | 0.675 | 0.602 |
| 38 | Tarn@Cocures | 0.789 | 0.763 | 0.889 | 0.648 | 0.788 | 0.868 | 0.845 | 0.888 | 0.875 | 0.569 | 0.824 | 0.855 | 0.873 | 0.849 | 0.628 |
| 39 | Tarn@Millau | 0.875 | 0.941 | 0.979 | 0.872 | 0.976 | 0.946 | 0.930 | 0.983 | 0.970 | 0.985 | 0.940 | 0.928 | 0.984 | 0.962 | 0.979 |
| 40 | Tarn@Montbrun | 0.898 | 0.914 | 0.977 | 0.956 | 0.866 | 0.942 | 0.945 | 0.979 | 0.967 | 0.921 | 0.941 | 0.945 | 0.976 | 0.967 | 0.930 |
| 41 | Tarn@Pinet | 0.829 | 0.881 | 0.764 | 0.900 | 0.846 | 0.882 | 0.888 | 0.812 | 0.934 | 0.873 | 0.860 | 0.879 | 0.806 | 0.933 | 0.874 |
| 42 | Tarnon@Florac | 0.555 | 0.663 | 0.203 | 0.786 | 0.601 | 0.605 | 0.657 | 0.138 | 0.904 | 0.636 | 0.613 | 0.679 | -0.051 | 0.893 | 0.643 |
| 43 | Taurion@RocheTalamie | 0.827 | 0.682 | 0.481 | 0.830 | 0.688 | 0.803 | 0.436 | 0.135 | 0.856 | 0.817 | 0.785 | 0.336 | 0.310 | 0.843 | 0.639 |
| 44 | Tech@Reynes | 0.905 | 0.911 | 0.971 | 0.938 | 0.930 | 0.934 | 0.943 | 0.976 | 0.968 | 0.975 | 0.915 | 0.916 | 0.972 | 0.964 | 0.975 |
| 45 | Tet@Vinca | 0.931 | 0.907 | 0.913 | 0.962 | 0.918 | 0.957 | 0.920 | 0.951 | 0.948 | 0.964 | 0.933 | 0.885 | 0.935 | 0.944 | 0.967 |
| 46 | Tinee@PontLune | 0.825 | 0.843 | 0.794 | 0.868 | 0.869 | 0.853 | 0.887 | 0.886 | 0.927 | 0.906 | 0.857 | 0.879 | 0.888 | 0.924 | 0.927 |
| 47 | Truyere@Grandval | 0.572 | 0.578 | -0.018 | 0.667 | 0.584 | 0.673 | 0.695 | 0.584 | 0.770 | 0.574 | 0.719 | 0.674 | 0.520 | 0.770 | 0.636 |
| 48 | Ubaye@RocheRousse | 0.611 | 0.568 | 0.224 | 0.862 | 0.771 | 0.676 | 0.747 | 0.709 | 0.907 | 0.909 | 0.314 | 0.769 | 0.780 | 0.903 | 0.900 |
| 49 | Vence@Francheville | 0.756 | 0.846 | 0.868 | 0.906 | 0.935 | 0.955 | 0.869 | 0.963 | 0.964 | 0.971 | 0.957 | 0.899 | 0.953 | 0.967 | 0.965 |
| 50 | Vienne@Bussy | 0.745 | 0.897 | 0.832 | 0.947 | 0.941 | 0.958 | 0.910 | 0.928 | 0.968 | 0.977 | 0.946 | 0.940 | 0.918 | 0.975 | 0.976 |

**Table S5:** The values of the individual associated evaluation metrics (NSE(Q), NSE(Qsea), NSE(dQ), NSE(FDC) and NSE(Qlow)) for MORDOR V0, V1 and SD for all the catchments over the period P2

| ID | BV | MORDOR V0 | | | | | MORDOR V1 | | | | | MORDOR SD | | | | |
|---|---|---|---|---|---|---|---|---|---|---|---|---|---|---|---|---|
| | | NSE(Q) | NSE(Qsea) | NSE(dQ) | NSE(FDC) | NSE(Qlow) | NSE(Q) | NSE(Qsea) | NSE(dQ) | NSE(FDC) | NSE(Qlow) | NSE(Q) | NSE(Qsea) | NSE(dQ) | NSE(FDC) | NSE(Qlow) |
| 1 | Agout@Fraisse | 0.904 | 0.899 | 0.867 | 0.864 | 0.916 | 0.965 | 0.949 | 0.943 | 0.873 | 0.960 | 0.958 | 0.929 | 0.948 | 0.870 | 0.949 |
| 2 | Agout@LaRaviege | 0.927 | 0.950 | 0.937 | 0.788 | 0.908 | 0.961 | 0.977 | 0.965 | 0.780 | 0.955 | 0.957 | 0.973 | 0.964 | 0.905 | 0.952 |
| 3 | Allier@Poutes | 0.871 | 0.897 | 0.782 | 0.827 | 0.802 | 0.894 | 0.916 | 0.797 | 0.820 | 0.909 | 0.883 | 0.914 | 0.802 | 0.794 | 0.916 |
| 4 | Ardeche@Sauze | 0.562 | 0.737 | 0.426 | 0.539 | 0.591 | 0.647 | 0.704 | 0.477 | 0.451 | 0.787 | 0.656 | 0.709 | 0.506 | 0.215 | 0.788 |
| 5 | Arn@Taillades | 0.843 | 0.847 | 0.852 | 0.763 | 0.687 | 0.772 | 0.850 | 0.782 | 0.589 | 0.823 | 0.714 | 0.868 | 0.766 | 0.365 | 0.826 |
| 6 | Arve@Arthaz | 0.954 | 0.980 | 0.932 | 0.967 | 0.917 | 0.973 | 0.983 | 0.951 | 0.979 | 0.917 | 0.973 | 0.988 | 0.966 | 0.971 | 0.915 |
| 7 | Behine@LaPoutroie | 0.944 | 0.979 | 0.901 | 0.791 | 0.937 | 0.953 | 0.975 | 0.936 | 0.920 | 0.950 | 0.961 | 0.980 | 0.962 | 0.910 | 0.946 |
| 8 | Breze@Meyrueis | 0.845 | 0.893 | 0.797 | 0.854 | 0.796 | 0.947 | 0.925 | 0.813 | 0.880 | 0.890 | 0.939 | 0.925 | 0.847 | 0.917 | 0.888 |
| 9 | Bromme@Brommat | 0.393 | 0.660 | 0.538 | 0.263 | 0.356 | 0.862 | 0.562 | 0.586 | 0.030 | 0.595 | 0.870 | 0.509 | 0.611 | 0.365 | 0.593 |
| 10 | Ceze@Bagnols | 0.759 | 0.832 | 0.418 | 0.710 | 0.814 | 0.905 | 0.799 | 0.327 | 0.451 | 0.691 | 0.872 | 0.826 | 0.589 | 0.760 | 0.830 |
| 11 | Chassezac@SteMarguerite | 0.937 | 0.927 | 0.889 | 0.758 | 0.934 | 0.926 | 0.974 | 0.944 | 0.835 | 0.984 | 0.912 | 0.975 | 0.944 | 0.841 | 0.986 |
| 12 | Creuse@Age | 0.919 | 0.937 | 0.877 | 0.739 | 0.957 | 0.943 | 0.979 | 0.924 | 0.828 | 0.972 | 0.952 | 0.979 | 0.919 | 0.890 | 0.977 |
| 13 | Dordogne@Bort | 0.869 | 0.871 | 0.862 | 0.584 | 0.781 | 0.951 | 0.902 | 0.910 | 0.860 | 0.915 | 0.953 | 0.912 | 0.897 | 0.857 | 0.917 |
| 14 | Doubs@Brenet | 0.646 | 0.506 | 0.575 | 0.106 | -0.038 | 0.818 | 0.546 | 0.574 | 0.599 | 0.497 | 0.823 | 0.610 | 0.512 | 0.621 | 0.570 |
| 15 | Doubs@Neublans | 0.887 | 0.814 | 0.764 | 0.586 | 0.749 | 0.869 | 0.896 | 0.890 | 0.610 | 0.884 | 0.895 | 0.888 | 0.871 | 0.641 | 0.888 |
| 16 | Drac@Sautet | 0.928 | 0.910 | 0.905 | 0.880 | 0.847 | 0.954 | 0.953 | 0.949 | 0.898 | 0.896 | 0.956 | 0.950 | 0.954 | 0.893 | 0.874 |
| 17 | Durance@Clapiere | 0.946 | 0.953 | 0.906 | 0.901 | 0.815 | 0.970 | 0.980 | 0.973 | 0.949 | 0.938 | 0.966 | 0.978 | 0.964 | 0.946 | 0.907 |
| 18 | Eyrieux@Pontpierre | 0.883 | 0.840 | 0.805 | 0.706 | 0.771 | 0.911 | 0.872 | 0.860 | 0.846 | 0.924 | 0.917 | 0.850 | 0.881 | 0.866 | 0.920 |
| 19 | Gage@GageII | 0.646 | 0.654 | 0.558 | 0.318 | 0.542 | 0.800 | 0.609 | 0.806 | 0.757 | 0.803 | 0.802 | 0.538 | 0.810 | 0.766 | 0.781 |
| 20 | Gardon@Corbes | 0.825 | 0.846 | 0.811 | -0.121 | 0.816 | 0.753 | 0.866 | 0.836 | 0.745 | 0.772 | 0.829 | 0.865 | 0.882 | 0.824 | 0.818 |
| 21 | Gardon@Generargues | 0.930 | 0.925 | 0.917 | 0.967 | 0.891 | 0.950 | 0.980 | 0.947 | 0.979 | 0.916 | 0.952 | 0.977 | 0.947 | 0.977 | 0.904 |
| 22 | GaveEstaube@Gloriettes | 0.897 | 0.960 | 0.856 | 0.968 | 0.934 | 0.952 | 0.986 | 0.927 | 0.976 | 0.953 | 0.952 | 0.985 | 0.918 | 0.975 | 0.942 |
| 23 | Goussant@PontRolland | 0.784 | 0.858 | 0.897 | 0.849 | 0.683 | 0.869 | 0.934 | 0.853 | 0.883 | 0.846 | 0.857 | 0.936 | 0.886 | 0.891 | 0.854 |
| 24 | Loire@Basset | 0.293 | 0.656 | 0.642 | 0.603 | 0.054 | 0.596 | 0.762 | 0.621 | 0.683 | 0.574 | 0.598 | 0.739 | 0.666 | 0.726 | 0.604 |
| 25 | Loire@LaPalisse | 0.853 | 0.834 | 0.548 | 0.884 | 0.397 | 0.873 | 0.887 | 0.514 | 0.857 | 0.577 | 0.879 | 0.877 | 0.649 | 0.882 | 0.530 |
| 26 | Lot@Castelnau | 0.947 | 0.965 | 0.930 | 0.943 | 0.636 | 0.935 | 0.978 | 0.974 | 0.944 | 0.962 | 0.945 | 0.985 | 0.976 | 0.958 | 0.983 |
| 27 | Mimente@Florac | 0.937 | 0.939 | 0.941 | 0.902 | 0.486 | 0.938 | 0.958 | 0.977 | 0.927 | 0.850 | 0.939 | 0.968 | 0.976 | 0.958 | 0.926 |
| 28 | Montane@Eyrein | 0.886 | 0.889 | 0.794 | 0.774 | 0.679 | 0.883 | 0.896 | 0.920 | 0.896 | 0.801 | 0.880 | 0.902 | 0.901 | 0.923 | 0.813 |
| 29 | Oriege@Campauleil | 0.652 | 0.598 | 0.462 | 0.186 | 0.350 | 0.592 | 0.615 | 0.771 | 0.614 | 0.515 | 0.602 | 0.683 | 0.713 | 0.766 | 0.529 |
| 30 | Ouveze@Bedarrides | 0.868 | 0.808 | 0.833 | 0.771 | 0.278 | 0.804 | 0.885 | 0.883 | 0.773 | 0.750 | 0.815 | 0.881 | 0.894 | 0.796 | 0.815 |
| 31 | Rizzanese@Barrage | 0.917 | 0.976 | 0.921 | 0.905 | 0.934 | 0.887 | 0.971 | 0.934 | 0.959 | 0.972 | 0.866 | 0.990 | 0.928 | 0.953 | 0.970 |
| 32 | Romanche@Chambon | 0.865 | 0.847 | 0.924 | 0.808 | 0.957 | 0.918 | 0.817 | 0.974 | 0.915 | 0.982 | 0.890 | 0.922 | 0.974 | 0.915 | 0.979 |
| 33 | Roya@Breil | 0.811 | 0.838 | 0.873 | 0.783 | 0.829 | 0.896 | 0.904 | 0.918 | 0.833 | 0.859 | 0.886 | 0.905 | 0.922 | 0.856 | 0.869 |
| 34 | Salat@Roquefort | 0.284 | -0.057 | 0.547 | 0.353 | 0.543 | 0.648 | -0.135 | 0.694 | 0.532 | 0.479 | 0.635 | 0.156 | 0.702 | 0.586 | 0.435 |
| 35 | Sioule@Fades | 0.847 | 0.684 | 0.889 | 0.452 | 0.869 | 0.617 | 0.760 | 0.891 | 0.766 | 0.883 | 0.546 | 0.784 | 0.894 | 0.547 | 0.885 |
| 36 | Souloise@Infernet | 0.928 | 0.897 | 0.982 | 0.930 | 0.970 | 0.928 | 0.964 | 0.983 | 0.919 | 0.978 | 0.912 | 0.961 | 0.975 | 0.911 | 0.986 |
| 37 | Stura@Lanzo | 0.909 | 0.936 | 0.974 | 0.900 | 0.910 | 0.939 | 0.943 | 0.980 | 0.961 | 0.920 | 0.933 | 0.942 | 0.975 | 0.956 | 0.937 |
| 38 | Tarn@Cocures | 0.851 | 0.927 | 0.883 | 0.848 | 0.799 | 0.907 | 0.906 | 0.904 | 0.936 | 0.896 | 0.902 | 0.901 | 0.908 | 0.926 | 0.901 |
| 39 | Tarn@Millau | 0.632 | 0.793 | 0.508 | 0.494 | 0.081 | 0.719 | 0.729 | 0.599 | 0.750 | 0.536 | 0.733 | 0.721 | 0.594 | 0.734 | 0.612 |
| 40 | Tarn@Montbrun | 0.768 | 0.723 | 0.858 | 0.861 | 0.658 | 0.824 | 0.866 | 0.861 | 0.610 | 0.693 | 0.835 | 0.838 | 0.864 | 0.620 | 0.682 |
| 41 | Tarn@Pinet | 0.899 | 0.939 | 0.968 | 0.926 | 0.946 | 0.910 | 0.927 | 0.970 | 0.956 | 0.956 | 0.876 | 0.921 | 0.977 | 0.955 | 0.937 |
| 42 | Tarnon@Florac | 0.923 | 0.898 | 0.936 | 0.897 | 0.903 | 0.939 | 0.914 | 0.925 | 0.971 | 0.934 | 0.922 | 0.891 | 0.919 | 0.970 | 0.919 |
| 43 | Taurion@RocheTalamie | 0.875 | 0.867 | 0.804 | 0.901 | 0.808 | 0.870 | 0.883 | 0.809 | 0.924 | 0.926 | 0.858 | 0.862 | 0.785 | 0.917 | 0.926 |
| 44 | Tech@Reynes | 0.596 | 0.631 | 0.026 | 0.853 | 0.579 | 0.588 | 0.610 | 0.284 | 0.812 | 0.717 | 0.560 | 0.581 | 0.313 | 0.815 | 0.698 |
| 45 | Tet@Vinca | 0.787 | 0.321 | 0.382 | 0.780 | 0.713 | 0.859 | 0.776 | 0.351 | 0.873 | 0.683 | 0.876 | 0.704 | 0.456 | 0.837 | 0.721 |
| 46 | Tinee@PontLune | 0.838 | 0.895 | 0.854 | 0.936 | 0.943 | 0.900 | 0.928 | 0.892 | 0.954 | 0.972 | 0.901 | 0.930 | 0.903 | 0.950 | 0.979 |
| 47 | Truyere@Grandval | 0.623 | 0.935 | 0.842 | 0.903 | 0.962 | 0.913 | 0.965 | 0.834 | 0.975 | 0.980 | 0.902 | 0.965 | 0.863 | 0.972 | 0.979 |
| 48 | Ubaye@RocheRousse | 0.707 | 0.759 | 0.823 | 0.895 | 0.872 | 0.945 | 0.811 | 0.936 | 0.924 | 0.868 | 0.951 | 0.844 | 0.927 | 0.931 | 0.864 |
| 49 | Vence@Francheville | 0.386 | 0.384 | 0.390 | 0.655 | 0.556 | 0.931 | 0.494 | 0.774 | 0.779 | 0.614 | 0.935 | 0.582 | 0.740 | 0.788 | 0.525 |
| 50 | Vienne@Bussy | 0.438 | 0.641 | 0.569 | 0.863 | 0.825 | 0.818 | 0.601 | 0.740 | 0.854 | 0.816 | 0.726 | 0.617 | 0.830 | 0.836 | 0.789 |

**Agout@Fraisse**

[Figure]

**Agout@Fraisse**

**Agout@Fraisse**

[Figure]

**Agout@LaRaviege**

[Figure]

**Agout@LaRaviege**

Year 1980

Year 1981

Year 1982

Year 1983

Year 1984

Year 1985

**Agout@LaRaviege**

[Figure]

**Allier@Poutes**

[Figure]

**Allier@Poutes**

[Figure]

**Allier@Poutes**

[Figure]

**Ardeche@Sauze**

[Figure]

**Ardeche@Sauze**

[Figure]

**Ardeche@Sauze**

[Figure]

Ardeche@Sauze

[Figure]

Ardeche@Sauze

**Ardeche@Sauze**

[Figure]

[Figure]

**Ardeche@Sauze**

**Ardeche@Sauze**

[Figure]

**Arn@Taillades**

[Figure]

**Arn@Taillades**

[Figure]

**Arn@Taillades**

**Arn@Taillades**

**Year 2006**

**Year 2007**

[Figure]

Arve@Arthaz

**Arve@Arthaz**

[Figure]

**Arve@Arthaz**

[Figure]

**Behine@LaPoutroie**

[Figure]

**Behine@LaPoutroie**

**Breze@Meyrueis**

Breze@Meyrueis

**Breze@Meyrueis**

[Figure]

**Breze@Meyrueis**

**Bromme@Brommat**

[Figure]

**Bromme@Brommat**

[Figure]

[Figure]

**Ceze@Bagnols**

**Ceze@Bagnols**

[Figure]

Chassezac@SteMarguerite

**Chassezac@SteMarguerite**

[Figure]

**Chassezac@SteMarguerite**

[Figure]

[Figure]

Chassezac@SteMarguerite

[Figure]

Chassezac@SteMarguerite

[Figure]

**Chassezac@SteMarguerite**

**Chassezac@SteMarguerite**

[Figure]

Creuse@Age

[Figure]

Creuse@Age

**Creuse@Age**

[Figure]

**Creuse@Age**

[Figure]

**Creuse@Age**

[Figure]

**Creuse@Age**

[Figure]

**Creuse@Age**

[Figure]

**Creuse@Age**

[Figure]

**Dordogne@Bort**

[Figure]

**Dordogne@Bort**

[Figure]

**Dordogne@Bort**

[Figure]

[Figure]

Dordogne@Bort

**Dordogne@Bort**

[Figure]

[Figure]

**Doubs@Brenet**

[Figure]

Doubs@Brenet

**Doubs@Brenet**

[Figure]

**Doubs@Brenet**

[Figure]

**Doubs@Brenet**

[Figure]

**Doubs@Brenet**

[Figure]

**Doubs@Brenet**

[Figure]

**Doubs@Neublans**

[Figure]

**Doubs@Neublans**

[Figure]

**Doubs@Neublans**

[Figure]

**Doubs@Neublans**

[Figure]

**Doubs@Neublans**

[Figure]

**Doubs@Neublans**

[Figure]

[Figure]

**Drac@Sautet**

**Drac@Sautet**

[Figure]

**Drac@Sautet**

[Figure]

[Figure]

Drac@Sautet

**Drac@Sautet**

[Figure]

**Durance@Clapiere**

[Figure]

**Durance@Clapiere**

[Figure]

[Figure]

Durance@Clapiere

**Durance@Clapiere**

[Figure]

**Eyrieux@Pontpierre**

[Figure]

**Eyrieux@Pontpierre**

**Eyrieux@Pontpierre**

[Figure]

Gage@GageII

**Gage@GageII**

[Figure]

[Figure]

**Gardon@Corbes**

[Figure]

**Gardon@Corbes**

[Figure]

[Figure]

Gardon@Corbes

**Gardon@Corbes**

**Year 2005**

**Year 2006**

**Year 2007**

**Gardon@Generargues**

[Figure]

[Figure]

**Gardon@Generargues**

**Year 1993**

**Year 1994**

**Year 1995**

**Year 1996**

**Year 1997**

**Year 1998**

**Gardon@Generargues**

[Figure]

Gardon@Generargues

[Figure]

GaveEstaube@Gloriettes

**GaveEstaube@Gloriettes**

**GaveEstaube@Gloriettes**

[Figure]

[Figure]

**Goussant@PontRolland**

**Goussant@PontRolland**

[Figure]

[Figure]

[Figure]

**Loire@Basset**

[Figure]

Loire@Basset

**Loire@Basset**

[Figure]

**Loire@Basset**

[Figure]

**Loire@Basset**

[Figure]

[Figure]

**Loire@Basset**

**Loire@Basset**

[Figure]

**Loire@Basset**

[Figure]

[Figure]

Loire@LaPalisse

**Loire@LaPalisse**

[Figure]

**Lot@Castelnau**

[Figure]

**Lot@Castelnau**

[Figure]

Lot@Castelnau

Lot@Castelnau

**Mimente@Florac**

[Figure]

**Mimente@Florac**

[Figure]

**Mimente@Florac**

[Figure]

**Mimente@Florac**

[Figure]

**Montane@Eyrein**

[Figure]

**Montane@Eyrein**

[Figure]

**Montane@Eyrein**

[Figure]

**Montane@Eyrein**

[Figure]

**Montane@Eyrein**

[Figure]

**Montane@Eyrein**

[Figure]

**Montane@Eyrein**

[Figure]

**Oriege@Campauleil**

[Figure]

**Oriege@Campauleil**

[Figure]

**Oriege@Campauleil**

[Figure]

**Oriege@Campauleil**

[Figure]

**Ouveze@Bedarrides**

[Figure]

**Ouveze@Bedarrides**

**Year 2003**

Obs
MORDOR V0
MORDOR V1
MORDOR SD

**Year 2004**

Obs
MORDOR V0
MORDOR V1
MORDOR SD

Rizzanese@Barrage

**Rizzanese@Barrage**

**Rizzanese@Barrage**

[Figure]

**Romanche@Chambon**

[Figure]

**Romanche@Chambon**

[Figure]

**Romanche@Chambon**

[Figure]

**Romanche@Chambon**

[Figure]

[Figure]

Roya@Breil

**Roya@Breil**

[Figure]

[Figure]

**Roya@Breil**

**Year 2000**

**Year 2001**

**Year 2002**

**Year 2003**

**Year 2004**

**Year 2005**

[Figure]

**Salat@Roquefort**

**Salat@Roquefort**

[Figure]

**Sioule@Fades**

[Figure]

**Sioule@Fades**

[Figure]

**Sioule@Fades**

[Figure]

**Sioule@Fades**

[Figure]

**Sioule@Fades**

[Figure]

**Sioule@Fades**

[Figure]

**Sioule@Fades**

[Figure]

**Sioule@Fades**

[Figure]

**Sioule@Fades**

[Figure]

**Souloise@Infernet**

[Figure]

**Souloise@Infernet**

[Figure]

**Souloise@Infernet**

[Figure]

**Stura@Lanzo**

[Figure]

[Figure]

**Stura@Lanzo**

**Stura@Lanzo**

[Figure]

**Tarn@Cocures**

[Figure]

**Tarn@Cocures**

[Figure]

**Tarn@Cocures**

[Figure]

**Tarn@Cocures**

**Year 2004**

[Figure]

[Figure]

**Tarn@Millau**

[Figure]

**Tarn@Millau**

[Figure]

[Figure]

**Tarn@Millau**

[Figure]

**Tarn@Millau**

[Figure]

Tarn@Montbrun

**Tarn@Montbrun**

[Figure]

[Figure]

**Tarn@Montbrun**

**Tarn@Montbrun**

[Figure]

**Tarn@Pinet**

[Figure]

[Figure]

Tarn@Pinet

**Tarn@Pinet**

[Figure]

**Tarnon@Florac**

[Figure]

**Tarnon@Florac**

[Figure]

[Figure]

Tarnon@Florac

**Tarnon@Florac**

[Figure]

**Year 2004**

**Year 2005**

**Year 2006**

**Taurion@RocheTalamie**

[Figure]

**Taurion@RocheTalamie**

[Figure]

[Figure]

Taurion@RocheTalamie

**Taurion@RocheTalamie**

[Figure]

**Taurion@RocheTalamie**

[Figure]

**Taurion@RocheTalamie**

[Figure]

**Taurion@RocheTalamie**

[Figure]

**Taurion@RocheTalamie**

**Year 2000**

**Year 2001**

**Year 2002**

**Tech@Reynes**

[Figure]

**Tech@Reynes**

[Figure]

**Tech@Reynes**

[Figure]

[Figure]

**Tet@Vinca**

[Figure]

Tet@Vinca

**Tet@Vinca**

[Figure]

**Tet@Vinca**

[Figure]

**Tinee@PontLune**

[Figure]

[Figure]

**Tinee@PontLune**

**Tinee@PontLune**

[Figure]

Tinee@PontLune

**Truyere@Grandval**

[Figure]

**Truyere@Grandval**

[Figure]

[Figure]

Truyere@Grandval

**Truyere@Grandval**

[Figure]

**Truyere@Grandval**

[Figure]

**Truyere@Grandval**

**Year 2004**

**Year 2005**

**Year 2006**

[Figure]

**Ubaye@RocheRousse**

**Ubaye@RocheRousse**

**Ubaye@RocheRousse**

[Figure]

**Ubaye@RocheRousse**

[Figure]

**Vence@Francheville**

[Figure]

**Vence@Francheville**

**Year 1981**

**Year 1982**

**Year 1983**

**Year 1984**

**Year 1985**

**Year 1986**

**Vence@Francheville**

[Figure]

**Vence@Francheville**

[Figure]

**Vienne@Bussy**

[Figure]

[Figure]

**Vienne@Bussy**

**Vienne@Bussy**

[Figure]

**Vienne@Bussy**

---

## Author Comment (AC2) · 16 May 2017

**Detailed response to the comments of referee 2**

We want to thank M. Hrachowitz for his accurate and helpful review of our manuscript. In this author comment, we list how each of the remarks provided by the referee was addressed. The comments made by the referee will be referred as RC and printed in bold; the authors' comments and answers as AC.

**1 RC: The manuscript will benefit from being proof-read by a native English**

**speaker to reduce the number of typos and language errors (grammar, syntax and use of specific words/terms).**
AC: To answer your suggestion, the final manuscript will be checked by native English speaker.

**2 RC: It will be of tremendous help for the reader if the author provided tables of (a) the catchments used (including names, geographical positions, catchment areas, elevation range, slopes, annual P, annual potential E, annual Q, modelling time period, and time step (b) the parameters of each model, the associated symbols, units, prior distributions (are these the same for all catchments?) and descriptions (c) all model components (i.e. states and fluxes), including their symbols, dimensions and descriptions. This would make it much more convenient to follow the Appendix, in which many symbols are not clearly defined at this point. If deemed suitable, these tables can be provided as Supplementary Material.**
AC: We agree. We propose to add as Supplement Materials a specific section (S1) that presents more in details the dataset of the 50 catchments. Table S1 presents the main features of the catchments dataset, including name, geographical position, area, elevation range, slope, annual P, annual PET, annual Q, time step, modeling periods P1 and P2. Concerning model description and parameters we added a supplementary table in Appendix A which summarize MORDOR V1/SD free parameters, units, prior range (the same for all the catchments) and description. In addition we completed the description of model fluxes and states in Appendix A. Table 1 was also improved. On the other hand, concerning historical model version (MORDOR V0) we only added explicit references to existing publications which describe the model.

**3 RC: Section 3.2.2 will benefit from a clearer description of the different criteria. For example, it remains unclear what is meant by "streamflow regime". I**

suppose it is the long-term seasonal pattern, but please make this more specific. Similarly, the cumulative distribution of flows is commonly referred to as flow-duration curve. A more consistent terminology will help the reader to better appreciate the manuscript. It is also not clear what is meant by 1st-lag flow derivative. Does this refer to the lag-1 autocorrelation? Of flows? Of the recession? Please elaborate!

AC: We agree. We reformulate section 3.2.2 as follows :

"The runoff signatures are viewed in such a way that streamflow data may be broken up into several samples, each of them a manifestation of catchment functioning (Euser et al., 2013; Hrachowitz et al., 2014; Westerberg and McMillan, 2015). Five different signatures are used in this study and are described in the following:

- time serie of flow is obviously the first signature which has to be reproduced by the model (hereafter called $Q$);

- long-term mean daily streamflow is used to focus on the capacity to reproduce seasonal variation of observations (hereafter called $Qsea$);

- flow duration curve focuses on the capacity to reproduce streamflow variance and extremes (hereafter called $FDC$);

- flow recessions during low flow period focuses on streamflow recessions (hereafter called $Qlow$);

- $lag-1$ streamflow variation is the last signature focusing on short term variability (hereafter called $dQ$ and computed as follows: $dQ(t) = Q(t) - Q(t-1)$).

To go further, model realism is also evaluated in regards to three other hydrological variables: (i) fractional snow cover ($FSC$); (ii) snow water equivalent ($SWE$); (iii) actual evapotranspiration ($ET$).

However, observations available for these variables suffer from many limitations and uncertainties (see section 2). Consequently, a specific evaluation is conducted and is explained in sections 4.2 and 4.3."

**4 RC: The post-calibration evaluation of the models with respect to snow and evaporation dynamics is an important point in this paper. Yet, no mention of this is made in section 3.2.2. How are MODIS data used to compare to model output? Spatial averages? What about the temporal resolution of the evaluation? Which performance metric was used? Some of this is mentioned later in the manuscript but I think this needs to be made clear in the methods section.**

AC: We agree. We propose to add a specific comment in section 3.2.2 to mention the other hydrological variables, see response to comment 3 above. In addition we added more details in sections 4.2 and 4.3 to clarify how the satellite data (MOD10 and MOD16) are used.

**5 RC: Related to (4), I did not understand how a fractional snow cover can be reproduced with lumped model formulations (VO and V1). This makes clearly sense for a semi-distributed model (Mordor SD). But obviously I missed something for the lumped versions. Please clarify!**

AC: For the lumped versions of the model, the fractional snow cover $FSC$ estimates are based on a statistical formulation founded on the hypsometric curve. More in details, during accumulation the $FSC$ is computed according to a monotonic crescent function as follows:

$$FSC(t) = 1 - \frac{\arctan \frac{\gamma(t)-fp1}{fp2} + fp3}{fp4} \tag{1}$$

where the parameters $fp1$, $fp2$, $fp3$ and $fp4$ vary from a catchment to an other as a function of the hypsometric curve and the orografic gradient. The state variable $\gamma(t)$

depends on the snow pack $S(t-1)$, its temperature $t_S(t-1)$, the snow $N(t)$ and its temperature $t_N(t)$ as follows :

$$\gamma(t) = \frac{S(t-1) \cdot t_S(t-1) + N(t) \cdot t_N(t)}{S(t-1) + N(t)} \qquad (2)$$

For melt, $FSC(t)$ depends on the previous $FSC$ and the evolution of the snow pack as follows :

$$FSC(t) = FSC(t-1) \cdot (\frac{S(t)}{S(t-1)})^{0.5} \qquad (3)$$

**6 RC: What is the reason behind using KGE for calibration (which is completely fine) but NSE for evaluation? Why is not the same metric used for both?**
AC: We use the KGE for calibration because of its good statistical properties, which are helpfull for parameters identification. On the other hand, model evaluation is based on NSE because this criterion is commonly used for evaluation of hydrological models and is therefore suitable to use as a benchmark for this study. In addition, it allows to consider different metrics for calibration and posterior evaluation.

**7 RC: The presentation of the results and discussion section would strongly benefit from a bit more detail. Detailed results are only shown for a few catchments with good overall performance. And even for these, it remains unclear how the modelled hydrograph looks like (in comparison to the observed one) and what the values of the individual associated calibration objective functions (i.e. the 3 individual KGEs) and evaluation metrics(the remaining criteria) are. In addition,I think it would also be valuable to show examples of catchments where the model adaptation did not work and also discuss why.**
AC: We propose to add as Supplement Materials two specific sections about the model comparison over calibration periods (S2) and over evaluation periods (S3). The section

S2 presents more in details the performance of MORDOR V0, V1 and SD over the calibration periods P1 and P2. For the three considered models, we show values of the individual associated calibration metrics (KGE(Q), KGE(Qsea) and KGE(FDC)) for all the catchments over the calibration periods P1 and P2. In addition we show, for each of the 50 catchments, the observed hydrographs and those modeled by MORDOR V0, V1 and SD (calibration mode). Similarly, section S3 presents more in details the performances of MORDOR V0, V1 and SD over the evaluation periods P1 and P2. For the three considered models, we show the values of the individual associated evaluation metrics (NSE(Q), NSE(Qsea), NSE(dQ), NSE(FDC) and NSE(Qlow)) for all the catchments over the evaluation periods P1 and P2. We show also, for each of the 50 catchments, the observed hydrographs and those modeled by MORDOR V0, V1 and SD (evaluation mode). Concerning snow and evapotranspiration processes, we extend the analysis to other catchments, with 8 nival catchments for $FSC$ and 8 pluvial catchments for $AET$.

**8 RC: Related to(7), it is mentioned that V1 provides substantial improvements compared to V0. As V1 is changed in various respects in comparison to V0, it would be great if the authors invested a bit of effort to analyze and document which part/adjustment of V1 contributes most to the improvement.**
Ac: Mordor V1 differs from V0 especially for water balance formulation and snow modelling. However is very difficult to trace the origin of the various improvements. Logically, for nival catchments the changes in snow modelling (and also the semi-distribution, see figure 4) are efficient. Concerning the water balance and so the improvement in the representation of evapotranspiration processes, we propose to add a specific comment in section 4.3 in order to analyze the origin of AET differences.

**9 RC: P.1,l.6: what is meant by "inflected"? Please rephrase.**
AC: Is has been changed in the revised manuscript.

**10 RC: P.1,l.8: should read as "...evapotranspiration estimates. The model comparison is...."**
AC: It has been changed in the revised manuscript.

**11 RC: P.1,l.22: should read as "...semi-distributed..."**
AC: It has been changed in the revised manuscript.

**12 RC: P.1,l.23: Nijzink et al. 2016 would fit in nicely here.**
AC: Good suggestion. We added the reference in the text.

**13 RC: P.1,l.23: what is meant by "To overpass hydrological singularity...."? Please rephrase.**
AC: It has been changed in the revised manuscript.

**14 RC: P.2,l.8: I may be worth referring to Hrachowitz et al. (2014) here.**
AC: Good suggestion. This paper is well suited to this context and we added the reference in the text both in introduction and in section 3.2.2.

**15 RC: P.2,l.15: should read as "...framework on the MORDOR...."**
AC: It has been changed in the revised manuscript.

**16 RC: P.2,l.20-22: irrelevant. Can be condensed.**
AC: We agree. We removed this paragraph.

**17 RC: P.2,l.26: should read as "......mainly in the Alps (18 catchments), the**

**Pyrenees (5 catchments) and the Massif Central..."**
AC: It has been changed in the revised manuscript.

**18 RC: P.2,l.29: should read as "...hydrological conditions. The average area of the study catchments is..."**
AC: It has been changed in the revised manuscript.

**19 RC: P.3,Table 1: not clear if the 22/17/19 parameters are all calibration parameters, as it seems in the Appendix that some of them are fixed. Please clarify.**
AC: We tried to clarify this in Table 1. See comment to RC 2.

**20 RC: P.3,l.4: should read as "...1635 mm/yr. With regard to...."**
AC: It has been changed in the revised manuscript.

**21 RC: P.3,l.10: should read as "...sub-daily time steps..."**
AC: It has been changed in the revised manuscript.

**22 RC: P.3,l.11: what is meant by " the shape of local gauges"? Please clarify.**
AC: We agree. We propose a new formulation : "the hourly records of locals gauges are used to compute areal precipitation and temperature at 12-, 8- and 6-hours time step."

**23 RC: P.3,l.12: that is ok, but it should be underlined that these are not observations but modelled estimates which can be subject to considerable uncertainty.**
AC: We agree. We propose a new formulation: "It has be noticed that these data

are not observations but modelled estimates which can be subject to considerable uncertainty."

**24 RC: P.4,l.1-2: should read as "...for being affected by many..."**
AC: It has been changed in the revised manuscript.

**25 RC: P.4,l.5: should read as "...provides fractional snow cover..."**
AC: It has been changed in the revised manuscript.

**26 RC: P.4,l.5: please explain what "fractional snow cover" describes. Are these spatial fractions? If yes across the entire catchment? Across a pixel? Which value was used to compare the modelled values with?**
AC: The satellite MOD10 product provides gridded snow cover time-series. In this study we average the gridded values at catchment scale in order to compute a fractional snow cover. It has been rephrase in the revised manuscript.

**27 RC: P.4,l.15: should read as "...interconnected storages."**
AC: It has been changed in the revised manuscript.

**28 RC: P.4,l.15: what is meant by "continuously"? Please clarify.**
AC: With this phrase we want to emphasize that MORDOR is a continuous hydrological model and not a event-based model. We propose a new formulation: "Is is a continuous model that can be can be used with a time step ranging from hourly to daily."

**29 RC: P.5,l.1: No, what is required is a \*representative\* estimate of areal precipitation. The mean (or any other measure of central tendency) will average**

**out extremes, which will,due to the non-linear nature of your(or better: any meaningful hydrological model), result in biased results.**
AC: We agree, the term "mean" is improper in this context. It has been changed in the revised manuscript.

**30 RC: P.5,l.19: "(ii) snow modelling have to be improved..." reads awkward. Please rephrase.**
AC: We rephrased as follows: "...representation of snow processes have to be improved..."

**31 RC: P.5,l.31: should read as "...evapotranspiration, the model..."**
AC: It has been changed in the revised manuscript.

**32 RC: P.5,l.32: what is meant by "neutralized"??**
AC: We propose a new formulation as follows: "(i) a surface interception: net rainfall and evapotranspiration capacity are calculated from the subtraction of $MET$ from rainfall"

**33 RC: P.6,l.11: It is not clear which part of the system the ground-melt component represents. What exactly does it do? Please clarify.**
AC: The snow ground melt corresponds to the melting component coming from the ground heat flux, see for example (DeWalle, D. R., Rango, A.,2008). From experimental studies (e.g. Whitaker and Sugiyama, 2005), the ground-melt rates range form 0.5 to 1 mm/day.

**34 RC: P.6,l.30: that is fine, but please specify if the gradients are set to fixed values or if they are calibrated (similar to rainfall multipliers). Where do the**

**values (fixed or prior distributions) come from? Literature? Please provide references.**

AC: The orographic gradients are calibrated with a uniform prior distribution whose upper and lower limits come from the climatic reanalysis used in this study (Gottardi et al., 2012). See the revised version of table 2 and appendix A.

**35 RC: P.7,l.2,section 3.2: I would suggest to rearrange this section for a better flow and to start with the calibration approach, followed by the split sample test and the post evaluation criteria.**

AC: We agree. Section 3.2 has been rearranged considering your suggestion. See response to points RC 3 and RC 4.

**36 RC: P.7,l.5: does this mean that you end up with 2 parameter sets for each catchments? Is the following analysis then based on these 100 parameter sets (i.e. 2 for each catchment)? Please describe in more detail what you are doing.**

AC: Yes we do that as explained in the text, see section 3.2.1. For each catchment we have 2 sets of parameters ($\theta_1$ from P1 and $\theta_2$ from P2) and all the results and performances are calculated from the 100 (2*50) simulations.

**37 RC: P.8,l.1-2: this resembles an approach described by Gharari et al. (2013). It would be good to refer to that paper.**

AC: Good suggestion. We added the reference in the text.

**38 RC: P.8,l.4ff: please clearly separate between criteria that are used for calibration (i.e. q, reg and qlc) and those used for post-calibration evaluation (i.e. etg, dq, snow cover, evaporation).**

AC: We agree. Section 3.2 has been rearranged considering your suggestion. See

response to RC 3 and RC 4.

**39 RC: P.8,l.17, eq.1: should this not read "KGEqcl"?**

AC: It has been changed in the revised manuscript.

**40 RC: P.8,l.20: "Numerous applications if this OF..." please provide references.**

AC: With numerous applications we refer to industrial studies made at EDF in an operational context. In spite of this, this $OF$ is inspired from Paquet et al. (2013). We rephrased the paragraph in section 3.2.3.

**41 RC: P.8,l.29: Do the V1 and SD models in \*all\* catchments outperform V0 or is it just on average? Please provide some representative examples for both – cases of improvements and cases where V1 and SD did not result in improvements.**

AC: According to section S3 of Supplement Materials (table S4 and S5), we can observe that for 73% of cases Mordor SD performs better than V0 in regard to streamflow signature. The Agout at La Raviege catchment (period P1) is a good example of important improvement of SD model. On the other hand for Ubaye at RocheRousse catchment Mordor V0 (period P1) performs better than SD.

**42 RC:P.9,l.4: the improvement is obvious, but I struggle to see the "spectacular" improvement. In addition,"most" seems also a bit exaggerated here: reg,qcl and etg show only minor improvements, if any. Please tone the statement down a bit to actually reflect what we can see in the figures.**

AC: We agree. We qualified our statement as follows : "As a conclusion, the new formulation (V1) provides a significant improvement of performances, specially for $q$ and $dq$ signatures."

**43 RC: P.10, Figure 5: are the NSE values the NSE values of the snow cover? Please clarify. In addition, please make sure that \*all\* figure captions in the manuscript are stand-alone, i.e. that the reader can fully understand a figure only by reading its caption.**

AC: Yes. Captions of figures 5, 6,7 and 8 were completed in order to take into account your suggestion.

**44 RC: P.10,l.1: what is meant by "overpasses"? please rephrase.**

AC: We propose to change "overpasses" by "outclasses".

**45 RC: P.10,l.2: what is meant by "...the interest of the..."? please rephrase.**

AC: We propose to change "...the interest of the..." by "Therefore, the semi-distributed scheme clearly shows its added value for nival catchments.

**46 RC: P.11,figures 6,7: see (43)**

AC: See comments above.

**47 RC: P.12,l.2: should read as "...that cannot be..."**

AC: It has been changed in the revised manuscript.

**References**

DeWalle, D. R., Rango, A. (2008). Principles of snow hydrology. Cambridge University Press.

Whitaker, A. C., Sugiyama, H. (2005). Seasonal snowpack dynamics and runoff in a cool temperate forest: lysimeter experiment in Niigata, Japan. Hydrological Processes,

19(20), 4179-4200.